



# ICON-Sapphire: simulating the components of the Earth System and their interactions at kilometer and subkilometer scales

Cathy Hohenegger[1], Peter Korn[1], Leonidas Linardakis[1], René Redler[1], Reiner Schnur[1],
Panagiotis Adamidis[2], Jiawei Bao[1], Swantje Bastin[1], Milad Behravesh[1], Martin Bergemann[1,2],
Joachim Biercamp[2], Hendryk Bockelmann[2], Renate Brokopf[1], Nils Brüggemann[1,3], Lucas Casaroli[1],
Fatemeh Chegini[1], George Datseris[1], Monika Esch[1], Geet George[1], Marco Giorgetta[1], Oliver Gutjahr[1,3],
Helmuth Haak[1], Moritz Hanke[2], Tatiana Ilyina[1], Thomas Jahns[2], Johann Jungclaus[1], Marcel Kern[1],
Daniel Klocke[1], Lukas Kluft[1], Tobias Kölling[1], Luis Kornblueh[1], Sergey Kosukhin[1], Clarissa Kroll[1],
Junhong Lee[1], Thorsten Mauritsen[4], Carolin Mehlmann[1], Theresa Mieslinger[1], Ann Kristin Naumann[1,5],
Laura Paccini[1], Angel Peinado[1], Divya Sri Praturi[1], Dian Putrasahan[1], Sebastian Rast[1],
Thomas Riddick[1], Niklas Roeber[2], Hauke Schmidt[1], Uwe Schulzweida[1], Florian Schütte[1], Hans Segura[1],
Radomyra Shevchenko[1], Vikram Singh[1], Mia Specht[1], Claudia Christine Stephan[1], Jin-Song von
Storch[1,5], Raphaela Vogel[6], Christian Wengel[1], Marius Winkler[1], Florian Ziemen[2], Jochem Marotzke[1,5],
and Bjorn Stevens[1]

[1]Max Planck Institute for Meteorology, Hamburg, Germany

[2]Deutsches Klimarechenzentrum, Hamburg, Germany

[3]Institut für Meereskunde, Universität Hamburg, Hamburg, Germany

[4]Department of Meteorology, Stockholm University, Stockholm, Sweden

[5]Center for Earth System Research and Sustainability (CEN), Universität Hamburg, Hamburg, Germany

[6]LMD/IPSL, Sorbonne Université, CNRS, Paris, France

**Correspondence:** Cathy Hohenegger (cathy.hohenegger@mpimet.mpg.de)





**Abstract.** State-of-the-art Earth System models typically employ grid spacings of O(100 km), too coarse to explicitly resolve main drivers of the flow of energy and matter across the Earth System. In this paper, we present the new ICON-Sapphire model configuration, which targets a representation of the components of the Earth System and their interactions with a grid spacing of 10 km and finer. Through the use of selected simulation examples, we demonstrate that ICON-Sapphire can already now (i) be run coupled globally on seasonal time scales with a grid spacing of 5 km and on monthly time scales with a grid spacing of 2.5 km, (ii) resolve large eddies in the atmosphere using hectometer grid spacings on limited-area domains in atmosphere-only simulations, (iii) resolve submesoscale ocean eddies by using a global uniform grid of 1.25 km or a telescoping grid with a finest grid spacing of 530 m, the latter coupled to a uniform atmosphere and (iv) simulate biogeochemistry in an ocean-only simulation integrated for 4 years at 10 km. Comparison to observations of these various configurations reveals no obvious pitfall. The throughput of the coupled 5-km global simulation is 126 simulated days per day employing 21% of the latest machine of the German Climate Computing Center. Extrapolating from these results, multi-decadal global simulations including interactive carbon are now possible and short global simulations resolving large eddies in the atmosphere and submesoscale eddies in the ocean are within reach.

## 1 Introduction

Earth System Models (ESMs) have evolved over the years to become complex tools aiming at simulating the flow of energy and matter across the main components – ocean, land, atmosphere, cryosphere – of the Earth System, with their distinctive ability to simulate an interactive carbon cycle (Flato, 2011). ESMs for instance allow tracking how water evaporates from the ocean, precipitates over continents, boosts net primary productivity, returns as freshwater via runoff into the salty ocean, affects the spatial distribution of marine organisms, before reevaporating and restarting the cycle anew. Yet ESMs operate on grid spacings of O(100 km). In the atmosphere, this is too coarse to explicitly resolve the vertical transport of energy and water due to atmospheric convection. Convection is the main mechanism by which radiative energy is redistributed vertically and the main production mechanism of rain in the tropics (Stevens and Bony, 2013). In the ocean, a grid spacing of O(100 km) is too coarse to explicitly resolve mesoscale eddies. Mesoscale eddies control the uptake of heat and carbon by the deep ocean and change the simulated nature of the ocean, from a smooth laminar to a chaotic turbulent flow, structuring the large-scale circulation along the way (Hewitt et al., 2017). Moreover, a grid spacing of O(100 km) can only crudely represent the effect of the heterogeneous land surface, of the bathymetry, of coastal as well as ice shelves on the flow of energy and matter. A grid spacing of 10 km would at least permit deep convection in the atmosphere (Hohenegger et al., 2020), mesoscale eddies in the ocean ("eddy rich" model, see Hewitt et al., 2020) and capture much of Earth's heterogeneity. In this study, we present the new configuration of the ICOsahedral Nonhydrostatic (ICON) model, called ICON-Sapphire, developed at the Max Planck Institute for Meteorology for the simulation of the components of the Earth System and their interactions at kilometer and subkilometer scales, on global and regional domains.

The first climate models were developed to represent atmospheric processes on the global scale and were successively extended to incorporate more components of the Earth System (e.g. Phillips, 1956; Smagorinsky, 1963; Manabe et al., 1965;





GARP, 1975). ESMs participating in the last assessment report (AR6) can now represent dynamical ice sheets, terrestrial

and marine vegetation that grows and dies, atmospheric chemistry, carbon, nitrogen, sulfur and phosphorus cycles as well as more traditional physical processes that are at play in the atmosphere, at the land surface and in the ocean (see section 1.5 in Chen et al., 2021). Given the finiteness of computer resources, the increase in the degree of complexity has come at the cost of increases in resolution. The approximations imposed by the limited resolution are thought to be a reason for well-known biases. In the tropics, it still rains many hours too early, the locations of the rainbelts are misplaced and too little rain falls over

continents, with implications for the representation of the biosphere, which depends crucially on the precipitation distribution (Fiedler et al., 2020; Tian and Dong, 2020). Prominent biases persist in Sea Surface Temperature (SST), such as warm biases in the upwelling regions at the western coasts of continents, cold tongue biases in the tropical Atlantic and Pacific, as well as a subpolar cold bias in the North Atlantic (Keelye et al., 2012; Richter and Tokinaga, 2020). State-of-the-art ESMs not only struggle in capturing the mean spatial distribution of precipitation and surface temperature, but also in replicating their

associated extremes (Wehner et al., 2020). These biases limit confidence in regional and dynamical aspects of climate change (Shepherd, 2014; Lee et al., 2022). As several of these biases involve interactions between the components of the Earth System, resolving such biases may not be possible by increasing the resolution of just one component.

Flavors of atmospheric models operating at kilometer and subkilometer scales exist. Global atmosphere-only climate simulations at such resolutions have been pioneered by the Nonhydrostatic ICosahedral Atmospheric Model (NICAM) group (Satoh

et al., 2005; Tomita et al., 2005; Miura et al., 2007) and typically employ a grid spacing of 14 km, allowing multi-decadal integration periods. The recent DYnamics of the Atmospheric general circulation Modeled on Non-hydrostatic Domains (DYAMOND) intercomparison project demonstrated that global, kilometer-scale simulations on short time periods (40 days) are now possible with a range of atmospheric models, with a grid spacing as fine as 2.5 km (Stevens et al., 2019). Independent of these efforts, kilometer grid spacings have been used for regional climate modelling activities on multi-decadal time scales, on

domain sizes that have evolved from small Alpine regions (Grell et al., 2000; Hohenegger et al., 2008) to continents (Ban et al., 2014; Liu et al., 2017; Stratton et al., 2018). The feasibility of hectometer simulations, where not only deep convection but also shallow convection can be resolved explicitly, has been demonstrated for the monthly time scale and for domains the size of Germany (Stevens et al., 2020a). Experience gained from simulating the atmosphere at kilometer scales (see e.g. reviews by Prein et al., 2015; Satoh et al., 2019; Hohenegger and Klocke, 2020; Schär et al., 2020) has robustly shown improvements

in the representation of the diurnal cycle, organization and propagation of convective storms. Furthermore, a more realistic representation of extremes, orographic precipitation and blocking is obtained, and interactions between mesoscale circulation triggered by surface heterogeneity and convection become visible.

As with atmosphere-only models, ocean-only global simulations have been conducted, using grid spacings as fine as 1/48° for short time scales (e.g. Rocha et al., 2016; Qiu et al., 2018; Flexas et al., 2019). Coupled models typically employ grid

spacings of 1°, but high resolution ocean models with grid spacings of 1/10° (Haarsma et al., 2016) and even 1/12° (Hewitt et al., 2016) have been coupled to low-resolution atmospheres. Notably, Chang et al. (2020) conducted a 500-year simulation with a grid spacing of 0.1° for the ocean and 0.25° for the atmosphere. Through the use of limited or nested domains, ocean simulations that can explicitly resolve submesoscale eddies by making use of hectometer grid spacings have been conducted





for specific current systems (e.g. Capet et al., 2008; Gula et al., 2016; Jacobs et al., 2016; Chassignet and Xu, 2017; Trotta et al., 2017). Submesoscale eddies have a horizontal size of 100 m to 10 km. With the increase in computational power, such simulations have become more affordable and now allow for intercomparison exercises (Uchida et al., 2022). As reviewed by Hewitt et al. (2017) and Chassignet and Xu (2021), ocean models with a grid spacing of 1/10° and finer lead to a better representation of upwelling regions as well as to a more realistic positioning and penetration of boundary currents, features which lead to a reduction of associated SST biases. Such models also benefit from a better representation of the interactions between mesoscale motions and bathymetry, when channels and straits become resolvable. Furthermore, resolving ocean eddies reduces subsurface ocean model drift and affects the ocean heat budget, since mesoscale eddies act to transport heat upwards whereas the mean flow transports heat downwards (Wolfe et al., 2008; Morrison et al., 2013; Griffies et al., 2015; von Storch et al., 2016). Even submesoscale eddies are thought to be potentially climate relevant as they promote a re-stratification of the mixed layer, impacting atmosphere-ocean interactions (Fox-Kemper et al., 2011), and additionally have been shown to strengthen the mesoscale and large-scale circulation when present in models (e.g. Schubert et al., 2021).

All these developments have paved the way to the ultimate step: resolving the flow of energy and matter by both small-and large-scale phenomena, on climate time scales and across all components of the Earth System; in other words, explicitly resolving convective storms in the atmosphere, mesoscale eddies and marginally submesoscale eddies in the ocean, on global domains, with a surface of the Earth that retains most of its heterogeneity. Regional climate model simulations have shown that feedbacks between the land and the atmosphere can potentially be of distinct strength and sign in simulations that explicitly resolve convection compared to simulations that parameterize convection, although uncertainties remain due to the use of prescribed lateral boundary conditions (Hohenegger et al., 2009; Leutwyler et al., 2021). Similar may be true concerning interactions between the various components of the Earth System, making the development of kilometer-scale ESMs particularly interesting.

In this paper, we present the new ICON-Sapphire configuration. Sapphires are blue, like the blue end of the visible light spectrum associated with short wavelengths. Likewise, ICON-Sapphire is designed to resolve small spatial scales, with targeted grid spacings finer than 10 km. Moreover ICON-Sapphire can be employed as an ESM being able to simulate the biogeochemical processes, both in land and ocean, that influence the flow of carbon. To illustrate the intended abilities of ICON-Sapphire, we present results of key simulations: (i) a one year global coupled 5-km simulation and its counterpart integrated for 2 months at 2.5 km towards resolving the ocean-atmosphere system on multi-decadal time scales with deep convection and mesoscale ocean eddies; (ii) a hectometer-scale atmosphere-only simulation conducted on a limited domain to resolve large eddies in the atmosphere; (iii) a coupled global simulation with an ocean grid spacing of 12 km refined to 530 m over the North Atlantic, as well as an uncoupled global simulation with an ocean grid spacing of 1.25 km, both to resolve submesoscale eddies in the ocean; (iv) a 10-km 4-year global ocean-only simulation including ocean biogeochemistry towards demonstrating the ESM ability of ICON-Sapphire. The overall goal of the paper is threefolds: first to describe ICON-Sapphire, thus serving as reference for future, more comprehensive studies; second to show that such ultra-high resolution simulations are technically feasible; and third to investigate to which extent expected features of the Earth System can be reproduced using a configuration where most of the relevant climate processes is represented physically.



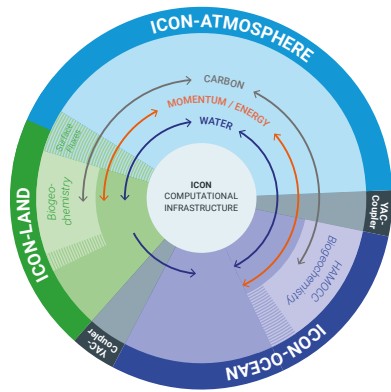

**Figure 1.** Overview of the components of the ICON-Sapphire configuration with their interactions.

It is generally thought that a throughput of 1 simulated year per day (SYPD), or at least not less than 100 simulated days
per day (SDPD) is needed for a climate model to be useful, meaning that simulations spanning many decades can be run
in a reasonable amount of time (Neumann et al., 2019; Schär et al., 2020). This has long prevented the use of kilometer grid
spacings. The use of Graphics Processing Units (GPUs), instead of the conventional Central Processing Units (CPUs), provides
a performance increase, bringing kilometer-scale simulations close to the target, as first demonstrated by Fuhrer et al. (2018) for
a near global, atmosphere-only, climate simulation. Giorgetta et al. (2022) ported the atmosphere version of ICON-Sapphire to
GPUs, making ICON-Sapphire versatile and well adapted to the new generation of supercomputers. This brings multi-decadal
ESM simulations at kilometer scales and global large-eddy simulations on short time scales within reach, as discussed in this
paper.

The outline of the paper is as follows. Section 2 describes the ICON-Sapphire configuration. This includes a description of its
main components, their coupling as well as I/O and workflow. Section 3 lists the throughput of different configurations. Section
4 illustrates key simulation examples, with an emphasis on the results from the fully coupled ICON-Sapphire configuration run
at 5 and 2.5 km. Section 5 summarizes the results and highlights some future development priorities.

## 2   Model description

The ICON model has originally been developed by the German Weather Service (DWD) for weather forecasts and by the
Max Planck Institute for Meteorology (MPI-M) for coarse-resolution (grid spacings of O(100 km)) climate simulations. The
DWD configuration, called ICON-NWP (Numerical Weather Prediction), only simulates the atmosphere together with a simple
representation of land surface processes, whereas the MPI-M configuration includes all components traditionally used in ESMs,
especially biogeochemistry in land and ocean. In the atmosphere, both model configurations share the same dynamical core
and tracer transport scheme but rely on distinct physical parameterizations. The ICON dynamical core is described in Zängl
et al. (2015) following the earlier work by Wan et al. (2013), whereas the coarse-resolution climate configuration of ICON,
called ICON-ESM, is presented in Jungclaus et al. (2021). Parallel to the development of this coarse-resolution configuration,



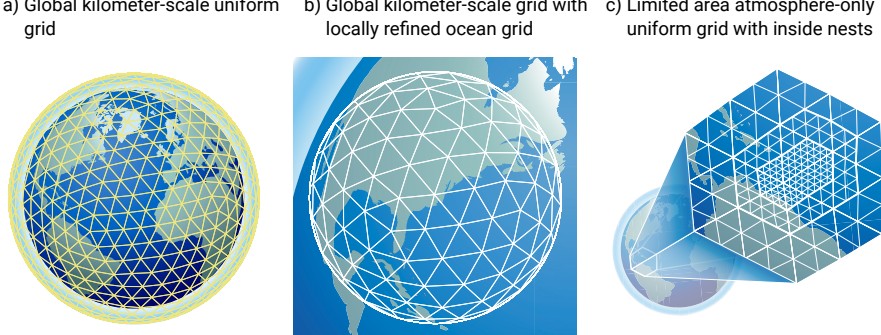

a) Global kilometer-scale uniform grid

b) Global kilometer-scale grid with locally refined ocean grid

c) Limited area atmosphere-only uniform grid with inside nests

**Figure 2.** Grid configurations supported by ICON-Sapphire: (a) global coupled kilometer-scale simulations with uniform grid in the atmosphere and ocean, (b) global coupled kilometer-scale simulations with uniform grid in the atmosphere and refined ocean grid over a specific region (telescope), with white atmosphere and blue ocean grid and (c) atmosphere-only large-eddy simulations over limited domains with the possibility of using inside nests to consecutively refine the resolution.

MPI-M has developed its new Sapphire configuration, targeting grid spacings of 10 km and finer, including the ability to be used as a true ESM with interactive carbon on multi-decadal time scales.

The Earth System in ICON-Sapphire is split into three main components, atmosphere, land and ocean with the ocean physical part including sea ice (see Fig. 1). Ocean biogeochemistry is simulated by the Hamburg Ocean Carbon Cycle (HAMOCC)
model, whereas all land surface processes, including biogeochemistry, are simulated by the Jena Scheme for Biosphere-Atmosphere Coupling in Hamburg (JSBACH). Interactions between the atmosphere and the ocean are handled by the Yet Another Coupler (YAC), whereas the land is implicitly coupled to the atmosphere via the turbulence routine. The biogeochemical tracers in the ocean are transported by the same numerical routines as for temperature and salinity, and carbon is exchanged with the atmosphere. The atmosphere component of ICON-Sapphire can be run globally or on limited domains, both support-
ing only uniform grids (see Fig. 2). The ocean component of ICON-Sapphire can only be run globally, but can make use of a telescope feature, which allows zooming in a region of interest to achieve higher resolution. These aspects, together with considerations concerning I/O and workflow, are presented in more detail below.

## 2.1 Atmosphere

The evolution of the atmosphere is simulated by solving the fully 3d nonhydrostatic version of the Navier-Stokes equations
and conservation laws for mass and thermal energy on the sphere, see Eqs. (3)-(6) in Zängl et al. (2015). The equations are integrated in time with a two-time-level predictor-corrector scheme. Due to the presence of sound waves, the dynamics is sub-stepped so that dynamic adjustments happen on temporal increments five-times smaller than physical adjustments, as schematically illustrated in Fig. 3. The spatial discretization is performed on an icosahedral-triangular C grid. On such a grid, the prognostic atmospheric variables are the horizontal velocity component normal to the triangle edges, the vertical wind





component on cell faces, the density, the virtual temperature and the full air density including liquid and solid hydrometeors.
The placing of these variables on the grid is illustrated in Fig. 4 of Giorgetta et al. (2018). The grid can be both global or
restricted to a limited area (see Fig. 2a,c). The limited area implementation follows Reinert et al. (2018). The configuration
uses a simple one-way or two-way nesting strategy at the boundary. Boundary conditions can be prescribed from larger-scale
simulations (or reanalyses) or set to double periodic, as recently employed in Lee and Hohenegger (2022) following Dipankar

et al. (2015). In addition, there is the possibility to use inside nests. In this case, all the simulations are integrated at the same
time but fine scales can be simulated over specific regions. In the vertical, a terrain following hybrid sigma z-coordinate (the
Smooth LEvel VErtical coordinate, SLEVE) is used (Leuenberger et al., 2010) with a Rayleigh damping layer in the top levels
after Klemp et al. (2008) using a Rayleigh damping coefficient of $1 \ \mathrm{s}^{-1}$.

The ICON-Sapphire configuration only retains parameterizations for the physical processes, which can unequivocally not be

represented by the underlying equations at kilometer scales. Those are: radiation, microphysics and turbulence. It may be ar-
gued that other processes, like shallow convection, subgrid-scale cloud cover and orographic drag, should still be parameterized
at kilometer scales (Frassoni et al., 2018). We nevertheless refrain from including further, or more complex parameterizations.
First, it is especially important to have a slim code base that can be easily ported on emerging new computer architectures to
fully exploit exascale computers given that the main roadblock preventing climate simulations at these fine grid spacings is

computer resource (Palmer and Stevens, 2019; Schär et al., 2020). Second, parameterizations generally do not converge to a
solution as the grid spacing is refined, unlike the basic model equations given appropriate discretization algorithms (Stevens
and Lenschow, 2001). Along similar lines, underresolved processes at kilometer scales may look closer to their resolved ver-
sion than what a parameterization may predict. Hohenegger et al. (2020) showed that many large-scale bulk properties of the
atmosphere, such as mean precipitation amount, already start to converge with a grid spacing of 5 km. Third, the need for

parameterizations depends upon the intended use of the model. For a weather forecast model, it is important to simulate the
atmospheric state as realistically as possible. In this sense, parameterizations may be viewed as a high-level way to tune a
model. But this is not the goal of the developed ICON-Sapphire configuration, which is used as a tool to understand the Earth
System and its susceptibility to change. In this context, the use of a minimalistic physics also helps understand which climate
properties depend upon details of the flow that remain underresolved or parameterized at kilometer scales.

The calling order of the three parameterizations is illustrated in Fig. 3 together with the coupling between dynamics and
physics. Radiation is parameterized by the Radiative Transfer for Energetics RRTM for General circulation model applications-
Parallel (RTE-RRTMGP) scheme (Pincus et al., 2019). It uses the two-stream, plane-parallel methods for solving the radiative
transfer equations. The RRTMGP part of the scheme employs a k-distribution for computing the optical properties based on
profiles of temperature, pressure and gas concentration. The RTE part of the scheme then computes the radiative fluxes using

the independent-column approximation in plane-parallel geometry. There are 16 bands in the longwave and 14 bands in the
shortwave and the spectrum is sampled using 16 g-points per band. The RTE-RRTMGP scheme can be employed both on GPU
and CPU architectures, unlike its predecessor PSrad (Pincus and Stevens, 2013). All simulations presented in this study were
nevertheless performed with PSrad as, at first, the RTE-RRTMGP scheme was unexpectedly slow on CPUs, an issue that has
now been solved. Differences between the two schemes are the use of a more recent spectroscopy and roughly twice as many



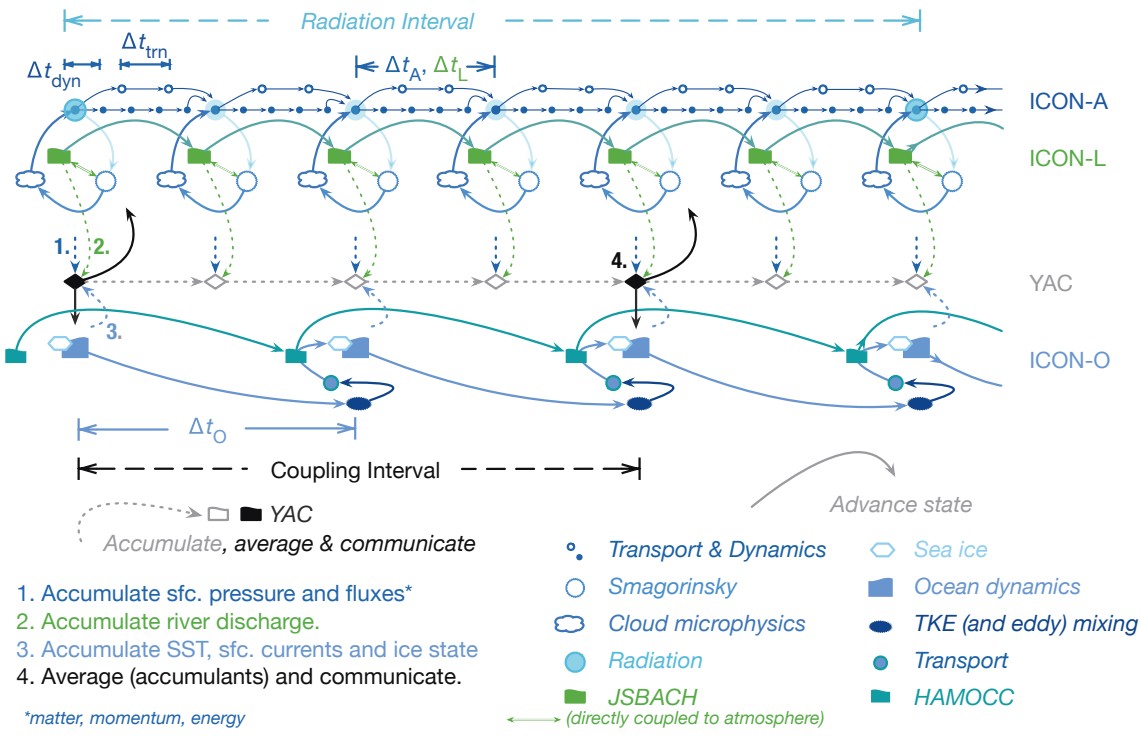

**Figure 3.** Time stepping in ICON-Sapphire.

g-points in RTE-RRTMGP, although a version with a reduced number of g-points, similar to the one used with PSrad, exists for
RTE-RRTMGP as well. Due to the high computational cost of radiation schemes in general, radiation is called less frequently
than the other parameterizations (see Fig. 3).

For the parameterization of microphysical processes, two schemes employed in the ICON-NWP configuration have been
implemented in ICON-Sapphire. There is the option to choose between a one-moment (Baldauf et al., 2011) and a two-moment
(Seifert and Beheng, 2006) scheme. All the simulations of this study were conducted using the one-moment scheme. The one-
moment scheme predicts the specific mass of water vapour, cloud water, rain, cloud ice, snow and graupel. In addition, the two-
moment scheme predicts the specific numbers of all hydrometeors and also includes hail as one supplementary hydrometeor
category. Saturation adjustment is called twice, before and after calling the microphysical scheme. All hydrometeors (number
and mass) are advected by the dynamics, but only cloud water and ice are mixed by the turbulence scheme and are optically
active. Condensation requires the full grid box to be saturated. ICON-Sapphire does not use a parameterization for subgrid-
scale clouds, leading to a binary cloud cover of 0 or 1 at each grid point.



Turbulence is handled by the Smagorinsky scheme (Smagorinsky, 1963) with modifications by Lilly (1962). The implementation and validation of the Smagorinsky scheme is described in Lee and Hohenegger (2022), following Dipankar et al. (2015). The Smagorinsky scheme is the scheme of reference for parameterizing turbulence in large-eddy simulations. Strictly speaking, it was not designed for applications at kilometer scales as it assumes that most of the turbulent eddies can be resolved explicitly by the flow (e.g. Bryan et al., 2003). It has nevertheless been successfully used with kilometer grid spacings, both in idealized and realistically configured simulations (e.g. Klemp and Wilhelmson, 1978; Langhans et al., 2012; Bretherton and Khairoutdinov, 2015). The surface fluxes are computed according to Louis (1979) using the necessary information provided by the land surface scheme, see section 2.2 for a description of the coupling between the land surface and the atmosphere.

## 2.2 Land

Land processes are simulated by the JSBACH land surface model version 4. It provides the lower boundary conditions for the atmosphere over land, namely albedo, roughness length and the necessary parameters to compute latent and sensible heat fluxes from similarity theory. The latter are part of solving the surface energy balance equation, which is, together with the multi soil thermal layers, implicitly coupled to the atmospheric diffusion equations for vertical turbulent transport following the Richtmyer and Morton numerical scheme. A multi-layer soil hydrology scheme is used for the prognostic computation of soil water storage (Hagemann and Stacke, 2015). Surface runoff and subsurface drainage are collected in rivers and are routed into the ocean by the hydrological discharge model of Hagemann and Dümenil (1997) with river directions determined by the steepest descent (Riddick, 2021). The physical, biogeophysical and biogeochemical processes included in JSBACH are described in Reick et al. (2021) for the JSBACH version 3.2. These processes are leaf phenology, dynamical vegetation, photosynthesis, carbon and nitrogen cycles, natural disturbances of vegetation by wind and fires, as well as natural and anthropogenic land cover change. New features in JSBACH 4 are the inclusion of freezing and melting of water in the soil, a multi-layer snow scheme (Ekici et al., 2014; de Vrese et al., 2021) and the possibility to compute the soil properties as a function of the soil water and organic matter content. Land surface processes are called at the same time as the main atmospheric processes (see Fig. 3) and integrated on the same grid. From an infrastructure point of view, JSBACH 4 has been newly designed in a Fortran2008 object-oriented, modular and flexible way.

Given the targeted horizontal resolution of ICON-Sapphire, there is no subgrid-scale heterogeneity in vegetation type. Only one vegetated tile is used, characterized by a vegetation ratio that gives the area coverage of vegetation over that tile, implicitly accounting for the presence of bare soil on the vegetated tile. A land cell can nevertheless be split between lake and vegetated or be fully covered by a glacier. Lakes are fractional but are not allowed in coastal cells. Surface temperature of lakes is computed by a simple mixed-layer scheme including ice and snow on lakes (Roeckner et al., 2003). In coastal regions, fractional land is permitted.

In this study, JSBACH has been used in a simplified configuration that only interactively simulates the physical land surface processes, as in weather forecasts or in traditional general circulation models. Leaf phenology is prescribed following a monthly climatology. Interactive leaf phenology has been employed in Lee and Hohenegger (2022) with idealized radiative convective equilibrium simulation conducted using a grid spacing of 2.5 km. Also, the hydrological discharge model was turned off.





Initialization of the hydrological discharge model reservoirs using remapped values taken from a prior MPI-ESM simulation failed. With this approach, unrealistically large discharge values occurred during the first three months, indicative of a poor initialization. Moreover, the discharge of a river is dumped into the uppermost ocean layer in a single cell, an unrealistic assumption with kilometer grid spacings. Preliminary studies are rather promising where the reservoirs are first spun up and
the discharge is distributed over multiple ocean cells, an improvement which will be implemented in the next model version to allow longer simulations.

## 2.3   Ocean

The ocean model ICON-O (see Korn, 2018; Korn et al., submitted) solves the hydrostatic Boussinesq equations, the classical set of dynamical equations for global ocean dynamics. The time stepping is performed using a semi-implicit Adams-Bashforth-
2 scheme in which the dynamics of the free surface is discretized implicitly. ICON-O evolves the state vector consisting of the horizontal velocity normal to the triangle edges, surface elevation, potential temperature and salinity. The UNESCO-80 formulation is employed as equation of state to compute density given potential temperature, salinity and depth. ICON-O uses, as the atmosphere, an icosahedral-triangular C grid. However, in contrast to the atmosphere, the global grid can be locally refined to create a "computational telescope" (Fig. 2b) that zooms into a region of interest. The numerics of ICON-O shares similarities
to the atmosphere component but has also important differences. Both components use a mimetic discretization of discrete differential operators but ICON-O uses the novel concept of Hilbert-Space Compatible Reconstructions to calculate volume and tracer fluxes on the staggered ICON grid (see Korn, 2018). The transport of the oceanic tracers potential temperature and salinity is performed using a flux-corrected transport scheme with a Zalesak limiter and a piecewise-parabolic reconstruction in the vertical direction, analogous to what is done for the atmospheric tracers. The numerical scheme of ICON-O allows for
generalized vertical coordinates (Singh and Korn, in preparation), of particular importance here are the depth-based $z$ and $z^*$-coordinates. In the $z$-coordinate, only the sea surface height varies in time. As the latter is added to the thickness of the first ocean layer, the combination of a thin layer at the surface with a strong reduction of sea surface height can lead to a negative layer thickness. This problem is avoided by using the $z^*$- coordinate. In this case, the changing domain of the ocean is mapped onto a fixed domain but with vertical coordinate surfaces that change in time.
Given the fine resolution of ICON-Sapphire, only a subset of the parameterizations available in ICON-O and described in detail in section 2.2 of Korn et al. (submitted) are used, namely a parameterization for vertical turbulent mixing and one for velocity dissipation. The parametrization of turbulent vertical mixing relies on a prognostic equation for turbulent kinetic energy and implements the closure suggested by Gaspar et al. (1990), where a mixing length approach for the vertical mixing coefficient for velocity and oceanic tracers is used. For velocity dissipation (or friction), either a "harmonic" Laplace, a "bihar-
monic" iterated Laplace operator, or a combination of both can be used. The viscosity parameter can be set to a constant value, scaled by geometric grid quantities such as edge length of triangular area, or computed in a flow-dependent fashion following the modified Leith closure in which the viscosity is determined from the modulus of vorticity and the modulus of divergence (Fox-Kemper and Menemenlis, 2008). The calling order between dynamics, physics and transport is illustrated in Fig. 3.



The sea ice model is part of ICON-O. It consists of a dynamic and a thermodynamic component. Sea-ice thermodynamics
describes freezing and melting by a single-category, zero-layer formulation (Semtner, 1976). The current sea ice dynamics are
based on the sea ice dynamics component of the Finite-Element Sea Ice Model (FESIM) (Danilov et al., 2015). The sea ice
model solves the momentum equation for sea ice with an elastic-viscous-plastic rheology. Since ICON-O and FESIM are using
different variable staggering, a wrapper is needed to transfer variables between the ICON-O grid and the sea ice dynamics
component, (see Korn et al., submitted). A new sea ice dynamics model has been developed to bypass these limitations (see
Mehlmann et al., 2021; Mehlmann and Korn, 2022) and will be employed in the future.

The ocean biogeochemistry component is provided by HAMOCC6 (Ilyina et al., 2013). It simulates at least 20 biogeo-
chemical tracers in the water column, following an extended nutrient, phytoplankton, zooplankton and detritus approach, also
including dissolved organic matter, as described in Six and Maier-Reimer (1996). It also simulates the upper sediment by 12
biologically active layers and a burial layer to represent the dissolution and decomposition of inorganic and organic matter as
well as the diffusion of pore water constituents. The co-limiting nutrients consist of phosphate, nitrate, silicate and iron. A fixed
stoichiometry for all organic compounds is assumed. Phytoplankton is represented by bulk phytoplankton and diazotrophs (ni-
trogen fixers). Particulate organic matter (POM) is produced by zooplankton grazing on bulk phytoplankton and enters the
detritus pool. Export production is separated explicitly into $CaCO_3$ and opal particles. The POM sinking speed is calculated
using the Microstructure, Multiscale, Mechanistic, Marine Aggregates in the Global Ocean ($M^4$AGO) scheme (Maerz et al.,
2020). Time stepping and horizontal grid are as in ICON-O.

## 2.4 Coupling

For the coupling between the atmosphere and the ocean (see Figs. 1 and 3), we use YAC (Hanke et al., 2016) version 2.4.2.
This version has been completely rewritten. In particular, each process now only has to provide its own local data to the coupler
without information about neighbouring processes. Furthermore, the internal work load is now evenly distributed among all
source and target processes.

The atmosphere provides to the ocean the zonal and meridional components of the wind-stress separately over ice and
over water, the surface freshwater flux as rain and snow over the whole grid cell and evaporation over the ocean fraction
of the cell, short- and longwave radiation and latent and sensible heat fluxes over the ocean, sea ice surface and bottom
melt potentials, the 10-m wind speed and sea level pressure. The ocean provides the sea surface temperature, the zonal and
meridional components of velocity at the sea surface as well as ice and snow thickness, and ice concentration. The interpolation
of the wind and velocity vector components is done using the simple 1-nearest-neighbor method when the grid spacing in the
ocean and atmosphere matches. By more complex geometries, this interpolation is done using Bernstein-Bézier polynomials
following Liu and Schumaker (1996). We use the so-called interpolation stack technique of YAC and apply a 4-nearest-neighbor
interpolation to fill target cells with incomplete Bernstein-Bézier interpolation stencils. All other fields are interpolated with
a first order conservative remapping. The land-sea mask is determined by the ocean grid. No correction is applied for global
conservation.





## 2.5  Input/Output

External data and initial conditions are interpolated onto the horizontal grid at a preprocessing step. Vertical interpolation is done online during the model simulation. Concerning the land surface, required external physical parameters are based on the
same data set (Hagemann, 2002; Hagemann and Stacke, 2015) as used in the past in the MPI-ESM (Mauritsen et al., 2019). These data sets have an original grid spacing of 0.5° x 0.5°. For experiments at kilometer scales, this is not optimal and work to derive these parameters directly from high-resolution data sets is under way. The orography is nevertheless derived from the Global Land One-km Base Elevation Project (GLOBE), which has a nominal resolution of 30". Likewise, the bathymetry has a 30" resolution, as provided by the Shuttle Radar Topography Mission (SRTM30_PLUS) data set (Becker et al., 2009). In the
atmosphere, for the simulations presented in this study, global mean concentration of greenhouse gases are set to their values of the simulated year, values taken from Meinshausen et al. (2017). Ozone varies spatially on a monthly time scale. The dataset is taken from the input datasets for Model Intercomparison Projects (input4MIPS), has a resolution of 1.875° by 2.5° (latxlon), and the year 2014 is chosen as default. Aerosols are specified from the climatology of Kinne (2019), which provides monthly data on a 1° grid.

Initial conditions for the atmosphere are derived from the European Centre for Medium-Range Weather Forecasts (ECMWF) operational analysis for the chosen start date. Initial conditions for soil moisture, soil and surface temperatures, as well as snow cover are also derived from the ECMWF analysis. In contrast, the ocean is spun up. Its initial state is taken from a spun-up ocean model run with climatological forcing. As the spinup methodology slightly differs between the different experiments presented in this study, it is described for each simulation individually in more detail in section 4 where the simulation results
are presented.

ICON-Sapphire uses asynchronous parallel input/output because efficient parallel I/O is the main performance bottleneck. Moreover, the size of a single 3d variable is too big to fit into the memory of one CPU. The model calculations are carried out in parallel and the output is distributed across the local memories of the cluster's CPUs. The employed approach is to hide the overhead introduced by the output behind the computation ("I/O forwarding"). By using dedicated I/O processes,
the simulations can be further integrated in time while dedicated CPUs handle the I/O. Moreover reading and writing files in parallel greatly reduces the time spent on I/O (see section 3).

One challenge of kilometer-scale simulations is the amount of output generated. At 2.5 km, the atmosphere contains 83886080 cells at each of the 90 vertical levels of our set-up, and the ocean 59359799 cells and 128 levels. The output has been mostly analyzed using Jupyter notebooks and Python, where efficient ways to analyze the output directly on the un-
structured grid have been developed. As part of ICON-Sapphire activities and of the next Generation Earth Modelling Systems (nextGEMS) project, an e-book has been developed as a community effort, providing information on the simulations as well as hints and example scripts. The e-book can be found at: easy.gems.dkrz.de. An important tool for selecting a subset of the output or in general for reformatting and transforming variables remains the Climate Data Operators (CDO), which is developed at MPI-M (Schulzweida, 2021). In order to reduce the memory requirement of CDO, single-precision 32-bit float arrays are now
kept in memory. This can reduce the memory requirement for memory-intensive operations by a factor of 2. New operators





have been created to support unstructured grids, namely: selcircle to select cells inside a circle; selregion to select horizontal regions; maskregion to mask horizontal regions; gh2hl to interpolate 3d geometric height to height levels; ap2pl to interpolate 3d atmosphere pressure to pressure levels, collmerge to generate a single global file out of the multifile restart patches and sellonlatbox, which now supports unstructured grids.

## 3 Computational throughput

Table 1 focuses on the performance of the global coupled simulations conducted with ICON-Sapphire as those are the type of simulations that ultimately should be integrated on multi-decadal time scales. In ICON terminology (Giorgetta et al., 2018), the considered grid spacings are R2B9 and R2B10, which approximately correspond to grid spacings of 5 and 2.5 km, respectively. Note that on the unstructured ICON grid, all cells have approximately the same area, and the grid spacing in kilometer corresponds to the square root of the mean cell area. The specific settings of the simulations are summarized in Table 2 and described in more detail in the next section when we present the simulation results. For our experiments, we write thirty-seven 2d atmospheric variables every 30 minutes and fourteen 3d variables every 3 hours. Nineteen land variables are written every 3 hours, whereas, for the ocean, 2d output is written every 1 h (23 variables) and 3 h (24 variables) and 3d output is written every 24 hours (42 variables). This gives a total of about 40 TB per month (uncompressed) using netCDF-4 and 32 bit as output format.

The production simulations have been performed on the supercomputer Levante of the German Climate Computing Center (DKRZ). Levante was ranked 86 in the top500 list in November 2021 (time of installation), and replaced the previous machine Mistral. Table 1 indicates that ICON-Sapphire scales very well with node number. Increasing the node number by a factor 6, from 100 to 600 nodes, leads to a factor 5.25 increase in simulated days per day (SDPD) for the 5-km grid spacing on Levante. The scaling starts breaking when using more than 400 nodes. Going from the old machine Mistral to the new machine Levante leads to a 5.6x increase in SDPD for the same number of nodes. There are two reasons for this increase. From a technical point of view, we expect a gain in throughput by roughly a factor 4 from the increase in the number of processors per node (128 on Levante versus 32 on Mistral). The additional gain is attributed to a more economic use of resources due to the introduction of asynchronous ocean output, which became available only recently.

Making use of these various advantages, Table 1 reveals that a throughput of 126 SDPD is achieved on Levante for a grid spacing of 5 km when using 600 nodes, or 21% of the compute partition. This is slightly larger than the minimum throughput of 100 SDPD considered as needed for climate simulations to be useful (see introduction). With a grid spacing of 2.5 km, a throughput of 20 SDPD is obtained on 600 nodes. On the one hand, it is slightly better than the theoretically expected 8 fold decrease in performance due to the doubling in grid spacing. On the other hand, it is presently still too small to allow conducting climate simulations at such a high resolution. At best, a throughput of 100 SDPD would be obtained if using the whole Levante and assuming a perfect scaling.

Instead of CPUs, GPUs can be used to obtain a better throughput. GPUs are available on the supercomputer JUWELS Booster at the Jülich Supercomputing Centre. It contains 936 compute nodes and one JUWELS Booster node is about 0.05





PFlops, compared to 0.003 PFlops for one Levante node in terms of their linpack benchmarks. Hence, in theory, we would

expect a performance increase of 17 going from Levante to JUWELS Booster, giving a throughput of almost 1SYPD for the 2.5-km configuration using 600 nodes. However, as shown in Giorgetta et al. (2022) by comparing atmosphere-only 5-km simulation conducted with ICON-Sapphire on JUWELS Booster (with GPU) and on Levante (with CPU), the performance increase is closer to 8 (see their Table 3). This would mean that 1 SYPD would be feasible with a grid spacing of 2.5 km using 1368 nodes of JUWELS Booster, so requiring about 1.5 times its size. In other words, global climate simulations with

ICON-Sapphire at a grid spacing of 2.5 km would be nowdays feasible on a 68 PFlops machine. Such a throughput would be possible already now on the new petascale Europe's most powerful supercomputer LUMI in Finnland, which has almost twice as many GPUs as JUWELS Booster and provides 150 PFlops. GPUs are not only interesting because of their performance increase but also because of their low energy consumption. JUWELS Booster gets about 50 PFlops per MW, whereas Levante gets about 4.6 PFlops per MW. So even if JUWELS Booster is half productive as expected from the number of PFlops per

node, it still does five times as much throughput per watt than Levante.

A grid spacing of 2.5 km is the highest grid spacing that has been currently experimented for the coupled ICON-Sapphire configuration. The atmosphere-only version of ICON-Sapphire has been run at 1.25 km (R2B11) on Levante on 908 nodes for 3 days, giving a throughput of 4 SDPD (see Table 1). Also, an ocean-only version of ICON-Sapphire has been run at 1.25 km on Levante (see Table 1). The throughput is 97 SDPD on 1024 nodes and 179 SDPD on 2048 nodes, 15% short of the expected

perfect scaling (194 SDPD on 2048 nodes). These numbers indicate that also the ocean begins to become a bottleneck to reach 1 SYPD with a grid spacing of 1.25 km. Although these numbers are by far too low to allow multi-decadal simulations, even on a machine like LUMI, seasonal time scales are in principle already possible on a global scale and with a grid spacing of 1.25 km using ICON-Sapphire.

**Table 1.** Performance of the two global coupled configurations run with a grid spacing of 5 km and 2.5 km, called G_AO_5km and G_AO_2.5km in Table 2 summarizing the setting of all simulations. The performance at 1.25 km is given for an atmosphere-only (A) and ocean-only (O) configuration. Under nodes, A stands for atmosphere and O for ocean. On Levante, the A/O numbers give the load balancing, on mistral it gives directly the number of processors. The specifications of Mistral and Levante nodes are as follows: Mistral node: 2x 18-core Intel Xeon E5-2695 v4 (Broadwell) 2.1GHz, connected with Fourteen Data Rate (FDR) Infiniband; Levante node: 2x AMD 7763 CPU; 128 cores.

| Grid spacing | Machine | Nodes | SDPD |
|---|---|---|---|
| 5 km | Mistral | 420 (300 A, 120 O) | 17 |
| " | Levante | 600, 24A:8O | 126 |
| " | Levante | 420, 24A:8O | 96 |
| " | Levante | 400, 24A:8O | 90 |
| " | Levante | 200, 24A:8O | 48 |
| " | Levante | 100, 24A:8O | 24 |
| 2.5 km | Levante | 600, 24A:8O | 20 |
| 1.25 km (A) | Levante | 908 | 4 |
| 1.25 km (O) | Levante | 1024 | 97 |
| " | Levante | 2048 | 179 |





**Table 2.** Summary of simulation settings. More information on the employed external data are in section 2.5 and on the ocean spinup in section 4 in the respective subsections. Abbreviations A, L, O and C for atmosphere, land, ocean and carbon, n/a for not applicable, $\Delta x$ for the horizontal grid spacing, $\Delta z$ for the vertical grid spacing, $N_z$ for the number of vertical levels, $H_{\mathrm{top}}$ for the domain top, $H_{\mathrm{bot}}$ for the ocean bottom, $H_{\mathrm{damp}}$ for the height at which damping starts in the atmosphere, $\Delta t$ for the time step. Forcing means the simulation providing the lateral boundary conditions in the limited-area, atmosphere-only simulations, and the simulation providing the atmospheric forcing in the ocean-only simulations. Given the strong vertical stretching in the upper atmosphere, the indicated $\Delta_z$ in the atmosphere is for layers below 14 km.

| Setting | G_AO_5km | G_AO_2.5km | R_A_620m | R_A_308m | G_O_1.25km | G_AO_tel | G_OC_10km |
|---|---|---|---|---|---|---|---|
| $\Delta x$: A and L | 5 km | 2.5 km | 620 m | 308 m | n/a | 5 km | n/a |
| $\Delta x$: O | 5 km | 5 km | n/a | n/a | 1.25 km | 0.53-12 km | 10 km |
| $N_z$: A | 90 | 90 | 90 | 90 | n/a | 90 | n/a |
| $N_z$: L | 5 | 5 | 5 | 5 | n/a | 5 | n/a |
| $N_z$: O | 128 | 128 | n/a | n/a | 112 | 112 | 128 |
| $\Delta z$: A | 25-400 m | 25-400 m | 25-645 m | 25-645 m | n/a | 25-400 m | n/a |
| $\Delta z$: L | 0.065-5700 m | 0.065-5700 m | 0.065-5700 m | 0.065-5700 m | n/a | 0.065-5700 m | n/a |
| $\Delta z$: O | 2-472 m | 2-472 m | n/a | n/a | 6-532 m | 6-532 m | 8-200 m |
| $H_{\mathrm{top}}$: A | 75 km | 75 km | 21 km | 21 km | n/a | 75 km | n/a |
| $H_{\mathrm{bot}}$: O | 5421 m | 5656 m | n/a | n/a | 5656 m | 5656 m | 6362 m |
| $H_{\mathrm{damp}}$ | 44 km | 44 km | 15 km | 15km | n/a | 44 km | n/a |
| $\Delta t$: A | 40 s | 20 s | 4 s | 2 s | n/a | 30 s | n/a |
| $\Delta t$: A rad. | 12 min | 12 min | 10 min | 10 min | n/a | 12 min | n/a |
| $\Delta t$: O | 80 s | 80 s | n/a | n/a | 45 s | 30 s | 600 s |
| $\Delta t$: AO | 12 min | 12 min | n/a | n/a | n/a | 12 min | n/a |
| Start date | 20.1.2020 | 20.1.2020 | 1.2.2020 18UTC | 1.2.2020 18UTC | 1.1.2020 | 20.1.2020 | 1.1.2013 |
| Length | 406 d | 71 d | 30 h | 30 h | 1 month | 90 d | 4 y |
| Ini state: A | ECMWF | ECMWF | ICON-NWP | ICON-NWP | n/a | ECMWF | n/a |
| Ini state: L | ECMWF | ECMWF | ICON-NWP | ICON-NWP | n/a | ECMWF | n/a |
| Ini state: O | spinup | spinup | n/a | n/a | spinup | spinup | spinup |
| Forcing | n/a | n/a | ICON-NWP | ICON-NWP | ERA5 | n/a | ERA5 |
| Vel. diss. | combi. | combi. | n/a | n/a | biharm. | biharm. | biharm. |
| Visc. par. | constant | constant | n/a | n/a | constant | constant | constant |
| YAC interp. | near neigh. | near neigh. | n/a | n/a | Ber.-Béz. | Ber.-Béz. | n/a |

## 4   Key simulation examples

In this section, we present key examples of simulations conducted with ICON-Sapphire to illustrate its intended abilities. The simulation settings are summarized in Table 2. As will be shown, the simulated fields agree to a satisfactory degree with expectations and observations. In addition, the versatility and good scalability of ICON-Sapphire allow its use in a variety of set-ups, at the forefront of exascale climate computing. Extrapolating from these results, ICON-Sapphire already now allows Earth System simulations with interactive carbon with a grid spacing of 10 km for multi-decadal time scales. At the other end

of the spectrum, ICON-Sapphire allows studying the effects of submesoscale eddies in the ocean, by employing its telescope feature, and large eddies in the atmosphere, by employing its limited-area ability, on the transport of energy and matter.





### 4.1 Simulating the coupled ocean-atmosphere system globally on seasonal time scales at 5 and 2.5 km

In this section, we present results from global coupled simulations conducted with ICON-Sapphire employing a uniform grid spacing of 5 km and 2.5 km, referred to as G_AO_5km and G_AO_2.5km for Global, Atmosphere-Ocean simulation with the suffix indicating the grid spacing (see Table 2 and Fig. 2a). The length of the simulation periods is unique, with G_AO_5km integrated for more than one year, from 20 January 2020 to 28 February 2021 and G_AO_2.5km for 72 days, from 20 January 2020 to 30 April 2020 to at least cover the time period of the new winter DYAMOND intercomparison study. The necessary initial and external data are listed in section 2.5. The ocean is spun up by first conducting an ocean-only simulation with a grid spacing of 10 km, initialized from the Polar science center Hydrographic Climatology 3.0 (PHC) observational data set (Steele et al., 2001) and run for 25 one-year cycles using the Ocean Model Intercomparison Project (OMIP) forcing by Röske (2006). Then the simulation is forced from 1948 to 1999 by the National Centers for Environmental Prediction (NCEP) reanalysis (Kalnay et al., 1996) and from 2000 to 2009 by the ECMWF reanalysis ERA5 (Hersbach et al., 2020). The obtained state serves as a new initial state for a 5-km ocean-only simulation forced by ERA5 and integrated over the time period 2010 to 20 January 2020. This end state is interpolated onto the 2.5-km grid to obtain the initial state of G_AO_2.5km. Through this initialization technique, the ocean is spun up and already entails mesoscale eddies, without having to use a multi-decadal spinup at 2.5 km, which is computationally too expensive. The obtained initial state also remains close to observations. The initial mean SST bias is 0.031 K with a standard deviation of 0.75 K, whereas the initial salinity bias amounts to -0.15 g kg$^{-1}$ with a standard deviation of 0.68 g kg$^{-1}$.

#### 4.1.1 5-km results

As ESMs aim to represent the flow of energy and matter in the Earth System, we start our analysis with energy. Figure 4 displays the seasonal cycle in the net top-of-the-atmosphere (TOA) energy. Too much energy leaves the Earth System in G_AO_5km during the first half of the year, by 3 W m$^{-2}$ to more than 6 W m$^{-2}$ compared to observations. The simulation adjusts with time and better matches the observations from September 2021 to February 2022 with a remaining underestimation between 1 and 3 W m$^{-2}$. Overall, this leads to a yearly imbalance of -4 W m$^{-2}$ in G_AO_5km. This imbalance results from a too high reflection of shortwave radiation, by 7.3 W m$^{-2}$, compensated in part by a correspondingly 3.3 W m$^{-2}$ too weak outgoing longwave radiation. The bias in shortwave radiation is due to too much reflection over oceans (11 W m$^{-2}$), whereas the land shows too little reflection (by 3.7 W m$^{-2}$). The differences are especially prominent over tropical oceans in regions of shallow convection and stratocumulus, which exhibit a too large cloud cover (not shown). This is expected from storm-resolving simulations whose grid spacings remain too coarse to properly resolve shallow convection (e.g. Hohenegger et al., 2020; Stevens et al., 2020a). The consequence is a spurious and strong increase of cloud cover the coarser the grid spacing is. In contrast, the bias in outgoing longwave radiation is more prominent over the extratropics, and especially over the northern hemisphere extratropics, with a deviation of 6 W m$^{-2}$ against 3 W m$^{-2}$ in the southern hemisphere extratropics and 2 W m$^{-2}$ in the tropics. The simulated too weak outgoing longwave radiation is in agreement with the northern hemisphere extratropics being much colder (and cloudier) than observations, a reflection of their larger land fraction (see further below).

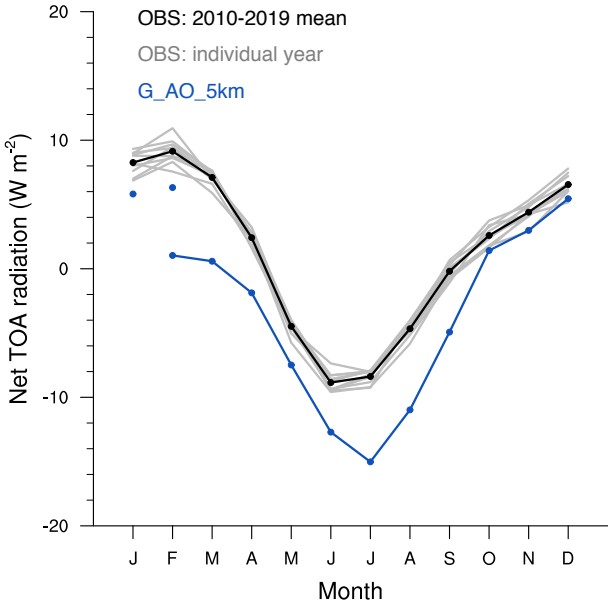

**Figure 4.** Seasonal cycle in net TOA radiation from observations and G_AO_5km. Observations from CERES-EBAF (Clouds and the Earth's Radiant Energy System-Energy Balanced and Filled) version 4.1 with a grid spacing of 1°. For G_AO_5km, the solid marked line shows the months in 2021 and the two dots in 2022.

As explained in Mauritsen et al. (2012), it is common practice in ESMs to tune the radiation balance to adjust uncertain parameters related to unresolved processes. Given the obtained yearly imbalance of -4 W m$^{-2}$, this need to adjust parameter values remains true at 5 km. It is not surprising given that ICON-Sapphire does not include a proper representation of shallow convection and thus also needs to include some factor to account for cloud heterogeneity, a factor that could be tuned (Mauritsen et al., submitted).

Given that the Earth System is losing too much energy, a cooling of the global mean surface temperature is expected. The mean values are 286.5 K in G_AO_5km against 287.9 K in observations. Biases are stronger over land than over oceans, with a mean bias of -1.7 K over land versus -0.6 K over oceans, likely reflecting the smaller heat capacity of the land surface (see also Fig. 5f in Mauritsen et al. (submitted) for an overview of the spatial distribution of temperature biases during JJA).

ICON-Sapphire explicitly resolves the modes of energy transport. In the atmosphere, extratropical storms (baroclinic eddies)
dominate the meridional energy transport, whereas atmospheric convection dominates the vertical one. We thus now investigate the representation of the large-scale circulation and of precipitation. Figure 5a,b shows meridional cross-sections of zonally averaged sea-level pressure in G_AO_5km and reanalysis for winter and summer. Except for a too low sea level pressure over Antarctica, the simulated sea level pressure lies within the internal variability from the reanalysis. The location and strength of the mid-latitude low-pressure systems and of the subtropical high-pressure systems match with ERA5 and their expected
seasonal migration and intensification is reproduced. A similarly good agreement is obtained by looking at the jets (see Fig. 5c-d). Both location and intensity of the jet in the winter hemisphere lie within internal variability. The position of the jet in



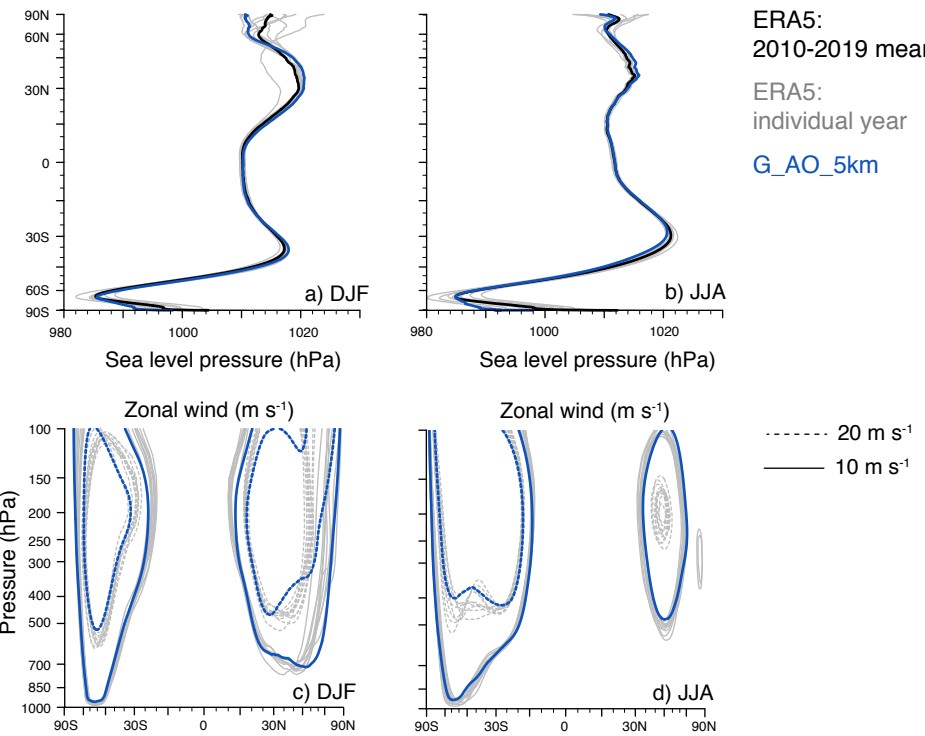

**Figure 5.** Meridional cross-sections of zonally averaged sea-level pressure (hPa) for (a) DJF and (b) JJA as well as height-latitude cross-sections of zonal wind velocity (m s$^{-1}$) for (c) DJA and (d) JJA from ERA5 (grid spacing of 30 km) and G_AO_5km. In G_AO_5km, DJF is the mean from December 2020 to February 2021.

the summer hemisphere is also well captured, but the simulated jet is weaker than the jet in the reanalysis, consistently so in the northern hemisphere.

Figure 6 shows the observed and simulated seasonal cycle in precipitation. The bands of enhanced precipitation within the extratropical storm tracks can be recognized together with their seasonal cycle. Averaged over the mid-latitudes, defined from 30° poleward, precipitation amounts to 2.37 mm day$^{-1}$ in G_AO_5km against 2.31 mm day$^{-1}$ for the 10-year observational mean and is larger than any of the observed yearly mean values, by at least 0.03 mm day$^{-1}$. Likewise, the migration of the tropical rainbelt is generally captured in the zonal mean (see Figs. 6a and c), although the equator stands out in G_AO_5km with too little precipitation. As shown in more detail by Segura et al. (in preparation) in their analysis of tropical precipitation

biases, this bias is especially visible in the Indo-Pacific region and linked to too low SSTs. Over land (Figs. 6b and d), in contrast, the seasonal migration of the tropical rainbelt is very well reproduced, but the amounts are overestimated. Tropical mean precipitation amounts over land are 3.6 mm day$^{-1}$ in G_AO_5km and 3.11 mm day$^{-1}$ in observationds for the 10-year mean, with no observed yearly value larger than 3.30 mm day$^{-1}$.

    Hence, it seems that ICON-Sapphire overestimates precipitation amounts. This is also true in the global mean, with an

overestimation by 0.4 mm day$^{-1}$. The overestimation is indicative of a too strong radiative cooling of the atmosphere. The

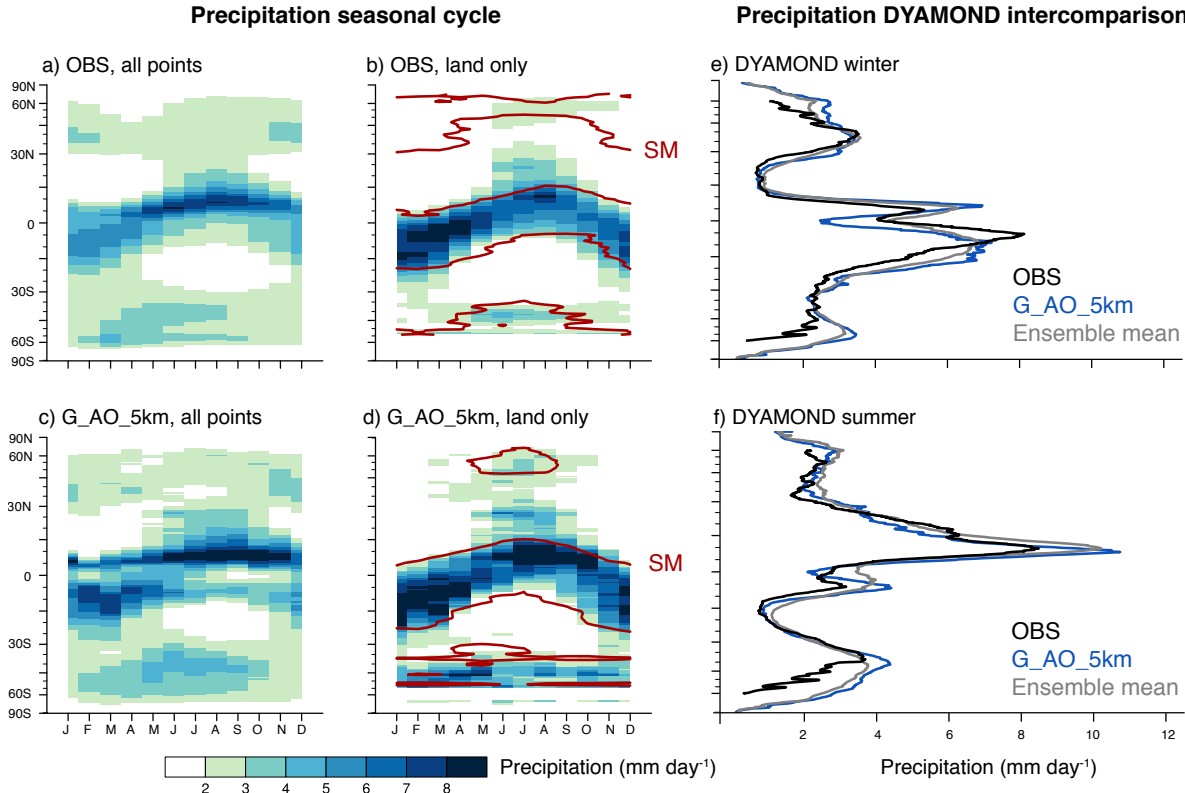

**Figure 6.** Seasonal evolution of zonal mean precipitation (shading) from (a-b) observations and (c-d) G_AO_5km. In (b,d), only land points are included with soil moisture (SM) contour overlaid in red (1.5 cm in (b) and 1 cm in (d) given their distinct magnitude). Observations are from the Global Precipitation Climatology Project (GPCP, grid spacing of 2.5°) and from the European Space Agency Climate Change Initiative (ESACCI) soil moisture (grid spacing of 0.25°) averaged over the years 2010-2019. For G_AO_5km the January and February values are from 2021. Panels (e-f) show the zonal mean precipitation from (e) the DYAMOND winter and (f) DYAMOND summer intercomparison study including the ensemble mean of participating models.

previously documented more negative net top-of-the-atmosphere energy budget in G_AO_5km than in observations (see Fig. 4) would indeed lead to an enhanced radiative cooling. However, even on the observational side, existing discrepancies between observed net radiation and observed precipitation suggest that precipitation amounts may in reality be larger than what observational precipitation datasets are suggesting (Stephens et al., 2012). Despite these apparent discrepancies, the simulated

precipitation is close to the mean precipitation as derived from the ensemble of storm-resolving models participating in the DYAMOND intercomparison study, as shown in Figs. 6e,f. As the DYAMOND models employ prescribed SST derived from ECMWF analysis, the good agreement gives us confidence in the behavior of our coupled model (G_AO_5km). The most noticeable difference, clearly out of the ensemble mean, is the underestimation of precipitation at the equator during DYAMOND winter.





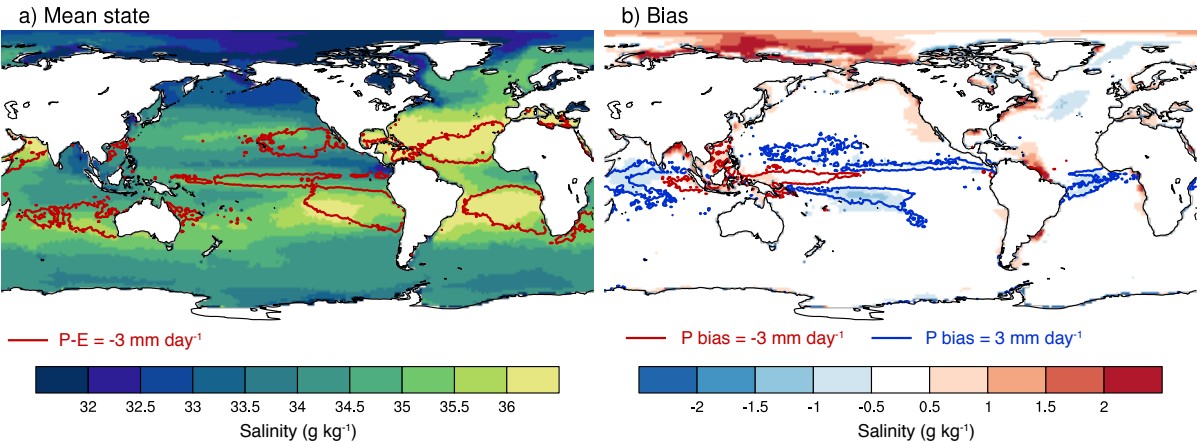

**Figure 7.** Salinity (g kg$^{-1}$) from G_AO_5km as (a) temporal mean and (b) bias relative to observations overlaid with contour lines representing P-E in (a) and precipitation bias in (b). Observations from PHC (grid spacing of 1°) for salinity and from GPCP for precipitation. For G_AO_5km, the time period 1 March 2020 to 28 February 2021 is considered.

Resolving convection is not only important for explicitly representing one of the main modes of energy transport in the tropics, but also for explicitly simulating the transport of water that is important for the biosphere. The brown contour line in Figs. 6b and d depicts the seasonal cycle in soil moisture. Not surprisingly, in the tropics, it matches very well with the migration of the tropical rainbelt, as also found in observations. In the northern extratropics, the moistening of the soil during summertime seems consistent with the simulated enhanced precipitation, but is inconsistent with observations. This inconsistency may be

related to the difficulty of measuring soil moisture during winter due to the presence of ice and snow. In any case, the hydrological budget is closed in ICON-Sapphire. As a second main discrepancy, soil moisture is consistently lower in G_AO_5km than in observations.

    The effect of precipitation on the ocean is investigated by comparing salinity and precipitation minus evaporation (P-E) patterns (Fig. 7). As expected, P-E imprints itself on salinity to a first order (Fig. 7a). Areas with evaporation much stronger

than precipitation are more salty. Biases in salinity are small with values smaller than 0.5 g kg$^{-1}$ except in the western Arctic. Interestingly, and even though we are comparing climatological to yearly values and observations from different sources, there seems to be a good correlation in the tropics between precipitation biases and salinity biases, especially in areas receiving too much precipitation. Positive localized salinity biases can also be recognized at the mouth of big rivers, like the Amazon, the Mississippi or the Congo, a signature of not having freshwater discharge into the salty ocean (see section 2.2).

The coupling between the ocean and the atmosphere is further investigated in Fig. 8 by computing correlations between SST, latent heat flux and precipitation. As explained in Wu et al. (2006), the correlation between heat flux and SST is positive when the SST leads and negative otherwise. G_AO_5km reproduces the order of magnitude of the correlation as well as its meridional variations, both between SST and latent heat flux and between SST and precipitation. The main bias is a positive correlation between SST and latent heat flux in the southern hemisphere subtropics, at odds with the observed negative corre-





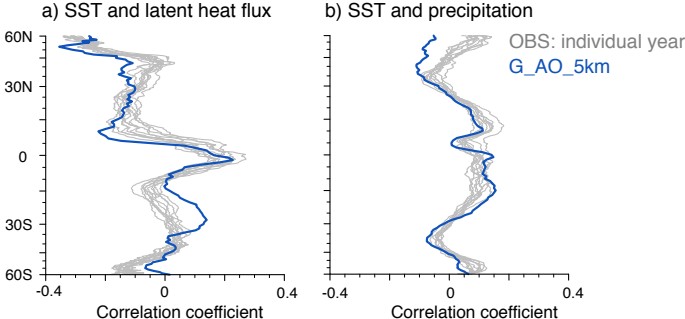

**Figure 8.** Meridional cross-section of zonally averaged correlation between SST and (a) latent heat flux and (b) precipitation. Observed SST from the Optimum Interpolation SST (OISST, grid spacing 0.25°), latent heat flux from the Objectively Analyzed air-sea flux (OAFlux_V3, grid spacing 1°) and precipitation from the Integrated Multi-satellitE Retrievals for Global Precipitation Measurement (IMERG, grid spacing 0.1°). Daily values are used.

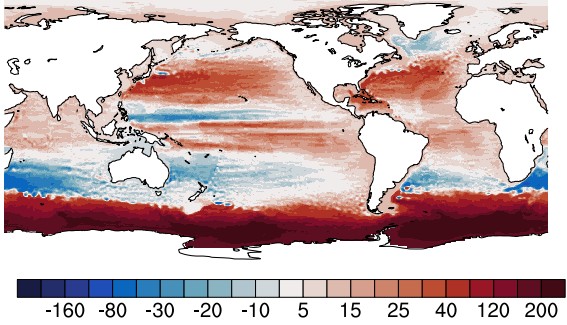

| Transport | G_AO_5km | OBS |
|---|---|---|
| Bering Strait | 1.2 | 1.1 |
| Caribbean Windward Passage | 4.0 | 3.8 |
| Drake Passage | 183.8 | 173.3 |
| Florida Bahamas Strait | 24.2 | 31.6 |
| Indonesian Throughflow | 14.9 | 13.0 |
| Mozambique Channel | 18.9 | 16.7 |
| Taiwan Luzon Straits | 2.4 | 2.4 |

**Figure 9.** Barotropic streamfunction (Sv) with associated major transports that can be computed from the streamfunction. Observations taken from Table J2 of Griffies et al. (2016), except for the Drake Passage which is taken from a more recent estimate (Donohue et al., 2016).

lation. Concerning precipitation, even though the strength of the simulated coupling between SST and precipitation is within observed individual years, it tends to fall either on the weak or on the strong side. The exception is in the higher northern latitudes that has a negative correlation bias between SST and precipitation, which is opposite to the positive correlation seen in observations.

     The dynamical coupling between the atmosphere and the ocean is reflected in the wind-driven circulation, which responds
fairly fast when coupling the ocean component to the atmosphere component. This circulation is described by the barotropic streamfunction (see Fig. 9) with positive values indicating clockwise flows. G_AO_5km simulates the main features of the wind-driven circulation: the subtropical and subpolar gyres in the North Atlantic and the strong circumpolar currents in the Southern Ocean, are captured by G_AO_5km. When considering the mass transports through major transects, the simulated transports compare reasonably well with those found in observations.

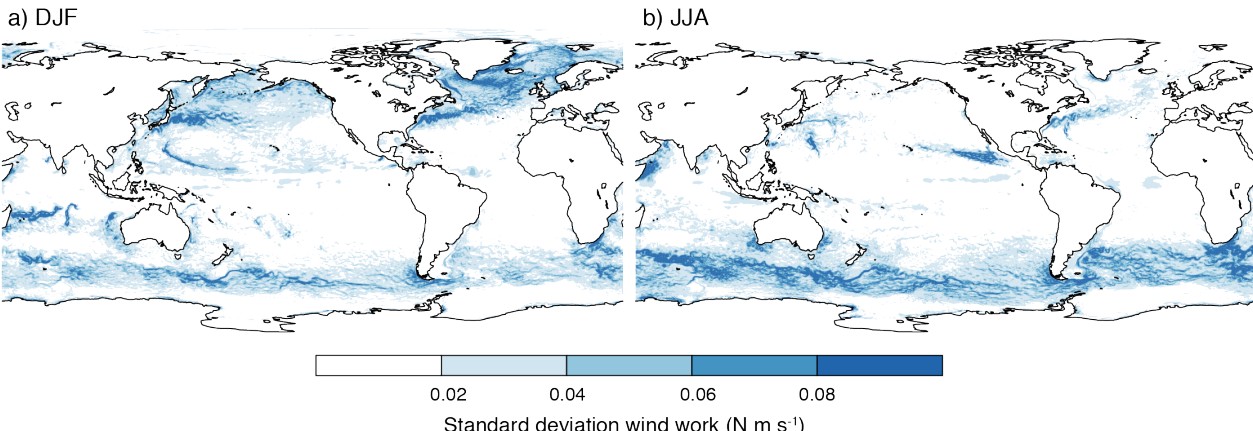

**Figure 10.** Standard deviation of wind work in G_AO_5km based on daily values for (a) DJF and (b) JJA.

The dynamical coupling between the atmosphere and the ocean also reflects itself in the variability of wind work (Fig. 10), which is defined as the product of surface wind stress and surface ocean currents (Wunsch, 1998). In eddy-rich regions such as the Gulf Stream, the Kuroshio current, the Agulhas Return Current, the Brazil-Malvinas confluence and in the Southern Ocean, strong dynamic interaction is seen year round, and is stronger in their respective winter hemisphere. Shift in direction associated with monsoonal winds induces the seasonal (boreal summer) Great Whirl off the Somalian coast that is also associated with

pronounced ocean-atmosphere interactions, which may affect the Indian summer monsoon. While this has been simulated in a regional coupled model (Seo et al., 2007; Seo, 2017), global storm-resolving configurations like G_AO_5km enable us to investigate even larger-scale effects. Tropical cyclone tracks are also visible in all major oceans. The total wind work displayed by Fig. 10 is 3.21 TW. In comparison, Flexas et al. (2019) estimated 4.22 TW for their $1/48°$ simulation.

As a final illustration of the coupling between the ocean and the atmosphere, we show in Fig. 11 three examples taken from

different regimes: a subsiding subtropical region, coincident with the area of operation of the EUREC[4]A (Elucidating the role of clouds-circulation coupling in climate) field campaign (Stevens et al., 2021), a region of deep convection with the Pacific InterTropical Convergence Zone (ITCZ), and the Gulf Stream (Fig. 11c), motivated by the findings of Minobe et al. (2008). G_AO_5km reproduces the observed diurnal cycle of net heat flux and temperature very well (Fig. 11a,b,c). The onset of ocean cooling and warming phases around 21 and 11 local time and the development of a diurnal warm layer during daytime are well

represented in G_AO_5km compared to the observed data. We nevertheless note that the simulated surface warming/cooling does not propagate far enough into the ocean, potentially pinpointing to too little mixing. The cross-section through the ITCZ (Fig. 11d,e) also reveals features that are well known from observations but can hardly be reproduced with coarse-resolution ESMs: a narrow region of about $5°$ of strong ascent around $6°N$ with associated boundary layer wind convergence and upper-level divergence at the top of the deep convection at around 300 hPa. Precipitation peaks below the convergence, a region which

coincides with high SST and strong SST gradients, two features favorable for the development of deep convection. Figure 11e also reveals the previously mentioned low SSTs and dip in precipitation at the equator. Finally, the atmospheric cross-section

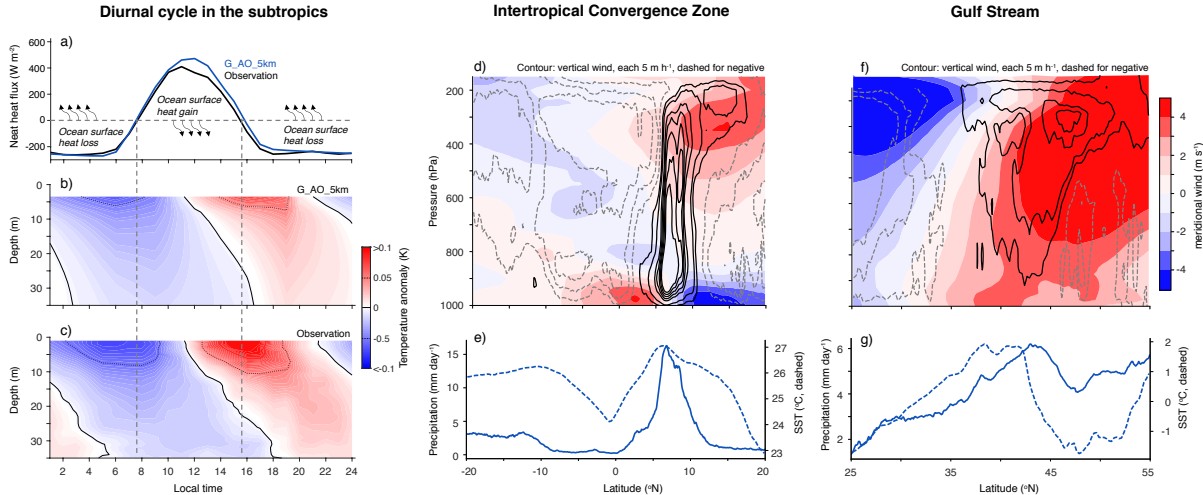

**Figure 11.** (a-c) Composite diurnal cycle of upper-ocean daily temperature anomaly in G_AO_5km as well as in observations with associated net heat flux at the ocean surface. Observations from ship data for the net heat flux and from three Slocum gliders for the temperature. Observations and G_AO_5km averaged between 57°W–58°W and 11.5°N–14.°N and from 21.1-29.2.2020. The remaining panels show meridional cross-sections from G_AO_5km averaged over February 2020 through (d-e) the central-eastern Pacific ITCZ and (f-g) atmosphere over the Gulf Stream. In (g) the SST equator-to-pole meridional gradient is removed. Zonal averages are taken between 150°W–90°W in (d-e) and 60°W–20°W in (f-g).

taken over the Gulf Stream (Fig. 11f,g) also reveals enhanced precipitation under the area of strong vertical motion. The latter sits on the top of the warm water of the Gulf Stream, albeit slightly shifted northward blending with the jet location.

### 4.1.2 2.5-km results

The global, ocean-atmosphere coupled simulation with a grid spacing of 2.5 km, G_AO_2.5km, employs the same set-up as G_AO_5km, except for its finer grid spacing and, accordingly, reduced time step (see Table 2). We now show four examples illustrating small-scale features embedded in larger-scale structures that such a model configuration aims at resolving. We also chose the four examples so as to illustrate interactions involving different components of the Earth System, rendered possible with the design of ICON-Sapphire.

Figure 12 focuses on the Southern Ocean. On the small scale, ocean eddies and filaments of enhanced velocity (Fig. 12d) imprint themselves on the salinity (Fig. 12a) and SST (Fig. 12c) fields, as expected. Superimposed on this variability, on the large scale, cold and salty water encounters warm and freshwater leading to the formation of a large-scale SST front. At this front, a low pressure system (Fig. 12b) develops with its associated circling wind (shown as streamlines), precipitation and clouds (Figs. 12e,f). But even in this large-scale low-pressure system, the precipitation is neither randomly nor uniformly 525 distributed, but aligns with the circling streamlines, forming mesoscale bands of precipitation. Comparing Figs. 12c and 12f, and perhaps surprisingly, a cooling of the surface below clouds as well as a freshening of the ocean below precipitation are not noticeable.

**Figure 12.** A low pressure system in the Southern Ocean and associated large- and small-scale features, visualized from G_AO_2.5km with (a) salinity, (b) salinity (shading) and sea level pressure (contour lines), (c) wind (streamlines) and surface temperature (shading), (d) wind (streamlines) and ocean velocity (shading), (e) wind (streamlines) and rain (blue shading), (f) wind (streamline) and rain (blue) and cloud (white). Field of view is towards Bay of Bengal. The tip of India and Sri Lanka can be recognized.

Figure 13 illustrates the breaking of tropical instability waves. Tropical instability waves develop already in ocean models with a grid spacing of 0.25° (Jochum et al., 2005); the finer grid spacing nevertheless allows resolving the submesoscale structure along the wave cusps with the development of secondary fronts. These secondary fronts rotate counterclockwise while the tropical instability waves break clockwise. Noteworthy is the imprint of these secondary fronts in the surface flux as shown in. Fig. 13b for latent heat.



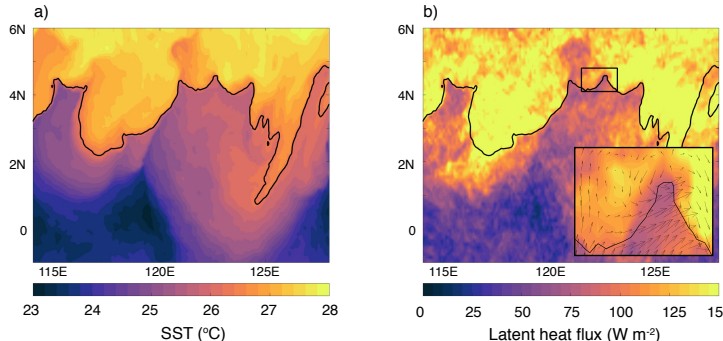

**Figure 13.** Breaking of a tropical instability wave in the Pacific in G_AO_2.5km with (a) SST and (b) latent heat flux. Black line shows the 26.3°C isoline. Snapshot for 5 February 2020 23:00.

In Fig. 14, the interactions between ocean, sea ice and atmosphere are illustrated. The passage of a polar low leads to the formation of narrow leads, a few kilometers wide, and of polynyas, compare Fig. 14c after the passage and Fig. 14a before the

passage. The sea-ice breakup results from the strong wind stress exerted by the southerly winds on the back of the polar low. By resolving these small-scale linear kinematic features in the sea ice, heat is released from leads and polynyas, whereas there is no or negative heat flux over thicker ice (compare Figs. 14c and d). Positive heat fluxes over leads and polynyas are to be expected and have been measured, for example, during the Surface Heat Budget of the Arctic Ocean (SHEBA) field experiment (Overland et al., 2000). Using a development version of ICON-Sapphire ran at 5 km, Gutjahr et al. (2022) also illustrated the

interactions between polar low, katabatic storms and dense water formation in the Irminger Sea.

As a last example, we confirm with Fig. 15 that G_AO_2.5km can resolve mesoscale circulations induced by surface heterogeneity and their impacts on convection, here due to the presence of islands. As expected, the differential heating between the land and the ocean leads to the formation of a mesoscale circulation (sea breeze), from the ocean into the land in the morning hours, which reverses in the evening hours. Precipitation bands develop at the edge of the circulation. These precipitation bands

not only form on the land during the day, but also over the ocean during night/early morning.

All in all, the results presented so far demonstrate the emergence of small-scale features, as intended by the use of ICON-Sapphire, and their interactions across components and scales of the Earth System. Even by only retaining a minimalistic physics in the atmosphere, made of radiation, microphysics and turbulence, the comparison to observations confirms that seasonal and large-scale features of the climate system are reproduced to a satisfactory level, without obviously wrong features.

## 4.2 Resolving large eddies in the atmosphere

Simulations like G_AO_5km and G_AO_2.5km enable studies of the effect of the small scales, in particular associated with deep convection and ocean mesoscale eddies, on the large-scale transport of matter and energy in the ocean-atmosphere system. On the atmosphere side, such grid spacings are unable to represent the effects of yet smaller scales, such as those associated with boundary layer turbulence or non-precipitating convection. A poor representation of non-precipitating clouds my lead to biases



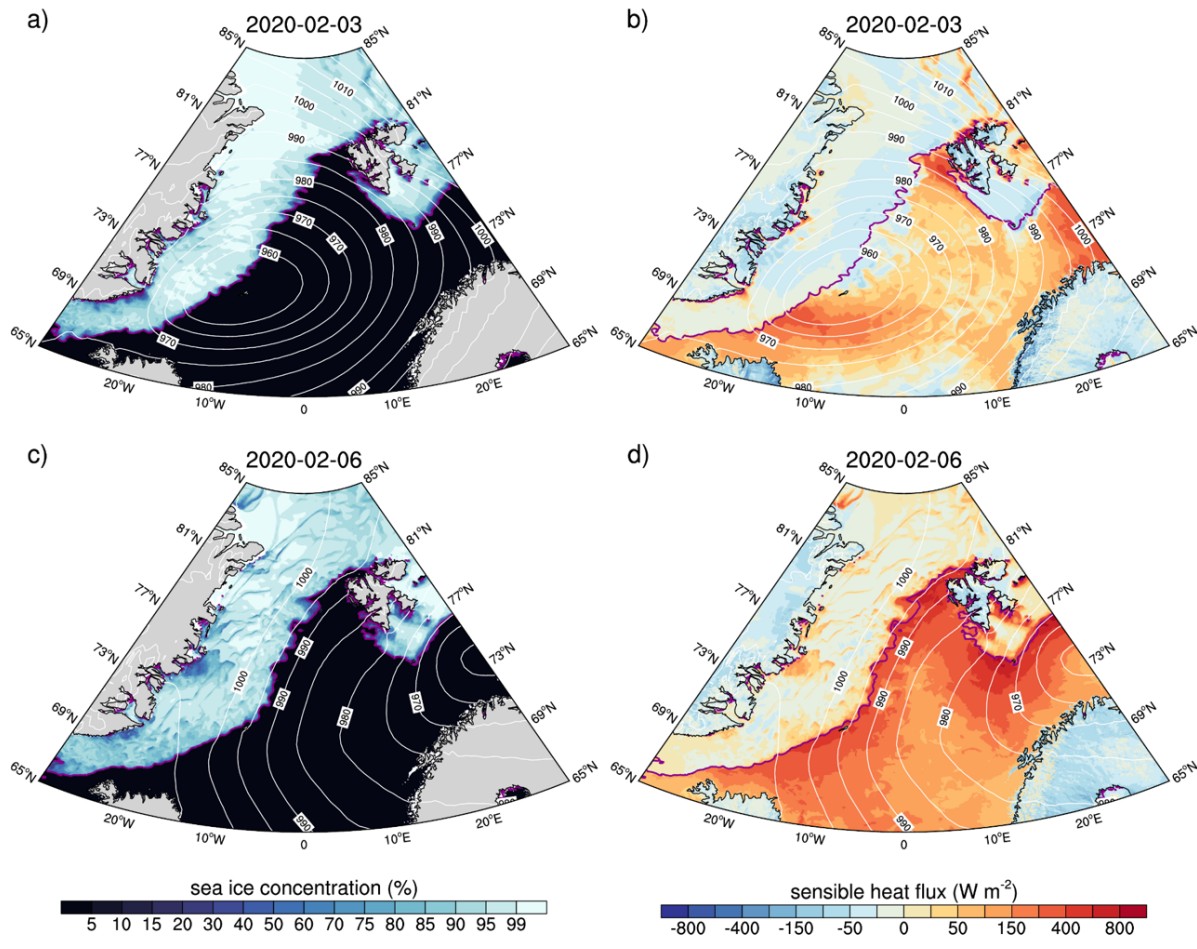

**Figure 14.** Sea ice breakup by a polar low and effect on sensible heat flux in G_AO_2.5km. Daily mean values are shown. White contour lines for sea level pressure (hPa), magenta line for the extent of the sea ice (15 % concentration).

in solar radiation, as documented in the previous section. Large eddies in the atmosphere can be resolved with ICON-Sapphire by using its limited-area and nesting capabilities (Fig. 2c), combined with hecto- and deca-meter grid spacings, as shown with the help of one example in this section. This makes it possible to perform large-eddy simulations with ICON-Sapphire for regional climate or process studies.

To demonstrate that ICON-Sapphire can be conducted over a limited area with an inside nest for local refinement, we
present the results of such a configuration (see Fig. 16 and Table 2). The chosen grid spacings are 620 m (R2B12) and 308 m (R2B13) for the outer and inner nests. The resulting two simulations are referred to as R_A_620m and R_A_308m for regional, atmosphere-only simulation with the grid spacing as suffix. The simulation domain extends over the subtropical western Atlantic, encompassing Barbados, to match the domain of operation of the EUREC[4]A field campaign (Stevens et al., 2021). It covers the area between 60°W–50°W and 10°N–16°N in R_A_620m and between 59.9°W–56°W and 12°N–14.5°N



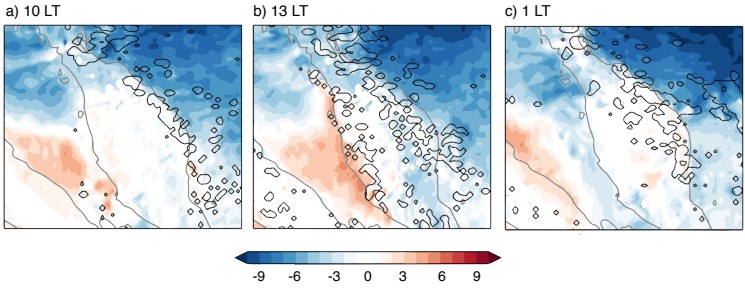

**Figure 15.** Interaction between sea breeze and convective precipitation in G_AO_2.5km around Malaysia on February 10 as a function of local time (LT). Contour (1 mm h$^{-1}$) for precipitation, shading for 10-m zonal wind velocity.

in R_A_308m. The simulations start 1 February 2020 21 UTC and are integrated until 3 February 0 UTC. We chose to simulate February 2 as that day shows different patterns of mesoscale cloud organization (Narenpitak et al., 2021). During the simulation period, the cloud field transitions from one form of mesoscale organization (Sugar) to another one (Flowers, following the terminology introduced by Stevens et al. (2020b)). The initial state and lateral boundary conditions are derived from a limited-area simulation conducted with the ICON-NWP model covering a large part of the tropical Atlantic, between

67°W–43°W and 0°–24°N, and using a grid spacing of 1.25 km. The lateral boundary conditions are read in every hour and two-way nesting is employed.

Figure 16 shows a snapshot of liquid water path and provides visual evidence of mesoscale organization of the cloud field, both in observations and reproduced in the two simulations. The observed cloud field is organized in separated clusters (Flowers) on the western half of the domain and in more fine grained patterns (Sugar) in the eastern half of the domain with equally

spaced lines. This organization is well reproduced by R_A_620m, and even the spacing between the Sugar lines seems to be similar to the observed one. Flowers are also simulated by R_A_308m. In both simulations, the cloud pattern seems more spotty than in observations. Averaged over the EUREC⁴A circle of flight operation, centered at 13.3°N, 57.717°W and with a diameter of 220 km, and during the 8-h flight time on February 2, the cloud cover amounts to 0.18 in R_A_620m and 0.17 in R_A_308m. In comparison, Konow et al. (2021) reported values between 0.1 and 0.2 from 5 out of 6 instruments with the

mean cloud cover of these 5 instruments around 0.15, close to the simulated values.

### 4.3   Resolving submesoscale eddies in the ocean

Not only in the atmosphere, but also in the ocean, ICON-Sapphire makes use of grid spacings at kilometer scales and below to allow for submesoscale dynamics. This capability is demonstrated by presenting the results of two types of simulations (see Table 2). One is a global ocean-only simulation with a grid spacing of 1.25 km, referred to as G_O_1.25km. In the second

type, the horizontal grid is refined up to 530m to create a "computational telescope" (Fig. 2b) that zooms into the North Atlantic, while the grid becomes increasingly coarser outside of this region down to 12 km. The later configuration, referred to as G_AO_tel, is run coupled to an atmosphere. The atmosphere employs a grid spacing of 5 km and the same settings



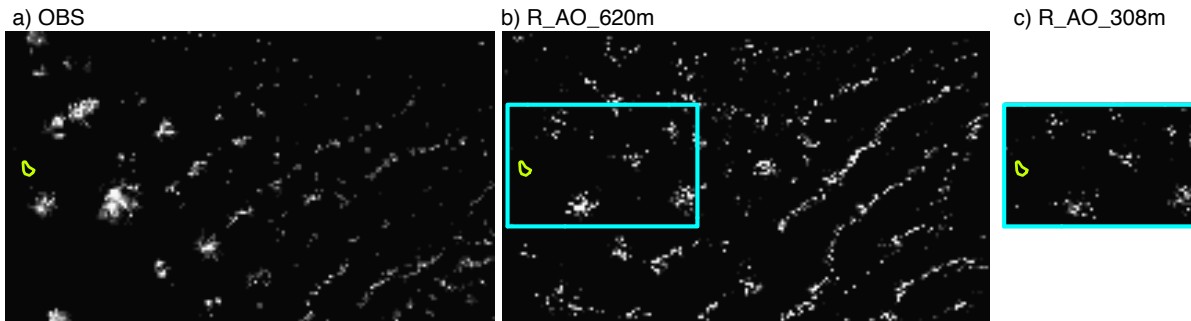

a) OBS          b) R_AO_620m          c) R_AO_308m

**Figure 16.** Snapshot of liquid water path on 2 February 2020 at 13 UTC from (a) observations, (b) R_AO_620m and (c) R_AO_308m. Observation from the Pathfinder ATMOSpheres (PATMOS), derived from GOES (grid spacing of 1 km).

and parameterizations as G_AO_5km. We choose the North Atlantic as focal region and start the simluation in boreal winter because of the favourable conditions for submesoscale dynamics.

The initial ocean state for the two simulations is obtained as follows. We branch off on 1.10.2019 from the 5-km ocean-only simulation that was used to spin up G_AO_5km (see section 4.1). That state is interpolated on a 2.5-km grid and ran until 1.1.2020, providing the initial state both for G_O_1.25km and for an ocean-only simulation using the same setting as G_AO_tel. After 20 days, the atmosphere is coupled to the ocean in the latter case.

     Figure 17 shows the details of the dynamics that can be simulated with a grid spacing of 1.25 km through a snapshot

of kinetic energy. In particular in the northern hemisphere, structures are visible that have spatial scales smaller than the first baroclinic deformation radius. The southern hemisphere shows less of these structures, which can be expected since submesoscale dynamics are thought to decrease during local summer conditions.

     One question to be answered within ICON-Sapphire is at which grid spacing we can expect submesoscale turbulence to develop. Here, we use the local Rossby number as a proxy for ageostrophic submesoscale dynamics and compare four different

grid spacings by making use of the simulation G_AO_5km, G_AO_2.5km, G_O_1.25km and G_AO_tel (Fig. 18). It becomes apparent that for grid spacings of 5 and 2.5 km, the Rossby number remains mostly smaller than one and only approaches one within the strong meanders of the Gulf Stream. In contrast, G_O_1.25km and G_AO_tel clearly show high Rossby numbers over much wider areas. In particular in G_AO_tel but also in G_O_1.25km, submesoscale eddies and fronts can be detected. Moreover, within the two simulations G_O_1.25km and G_AO_tel, not only submesoscale eddies but also the Gulf Stream

northern wall with narrow troughs and broad crests above which we observe reduced submesoscale activity can be seen. The flow features are similar to what has become familiar in simulations conducted on limited-area domains (see Fig. 21 in Chassignet and Xu (2017)). Given Fig. 18, we conclude that for our ICON-Sapphire configuration in the North Atlantic during boreal winter, the critical grid spacing to allow for submesoscale dynamics is somewhere between 2.5 and 1 km. Also, we conclude that the telescope feature works as intended and can even be used coupled to an atmosphere.



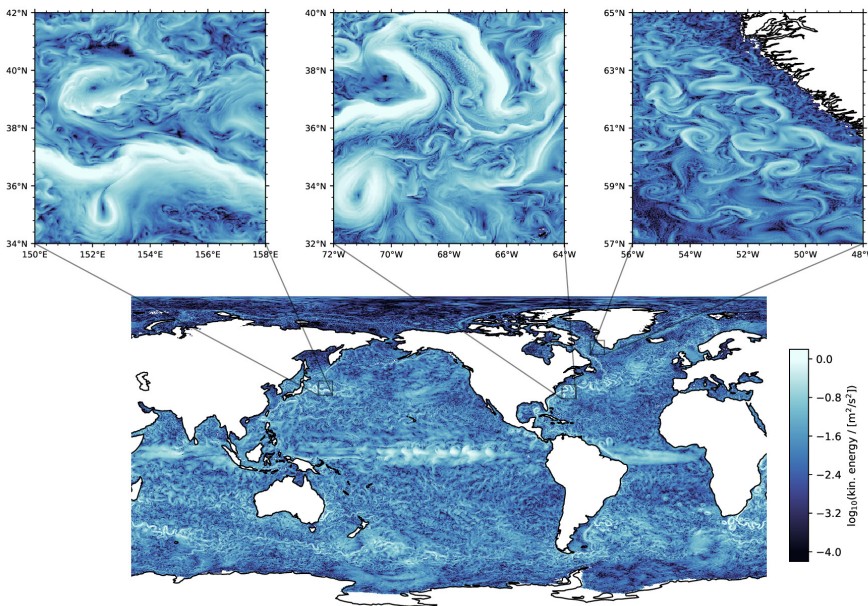

**Figure 17.** Snapshot of kinetic energy of G_O_1.25km with three additional zooms to highlight the richness of the dynamics that can be observed at this resolution.

### 4.4 ICON-Sapphire as an ESM

ICON-Sapphire includes all the components of an ESM, inherited through past modelling efforts at MPI-M. As a first step towards simulating interactively carbon at kilometer scales, we present in this section the results of a global ocean-only simulation run at 10 km (R2B8) with ocean biogeochemistry, one prerequisite for including an interactive carbon cycle. Phytoplankton induces radiative heating by absorbing shortwave radiation in the ocean. This in turn affects ocean physics as well as the carbon cycle and thus climate (Paulsen et al., 2018; Asselot et al., 2021). The presented simulation is referred to as G_OC_10km for global, ocean-only simulation with carbon and the suffix for the grid spacing (Table 2). It is run for 4 years, from 2013 to 2016, and forced by ERA5. To initialize the model, the physical ocean fields are taken from a spun-up ocean-only 10-km simulation. The initial biogeochemical fields are interpolated from a well spun up 40-km run. Results from the last year of the simulation (2016) are used for model evaluation. The simulated phytoplankton concentration is converted into chlorophyll-a concentration using a constant P/Chl ratio and compared to satellite observations from the Moderate Resolution Imaging Spectroradiometer MODIS-aqua (see Fig. 19). Given the maximum optical depth that satellite sensors can ultimately perceive (Gordon and McCluney, 1975), the simulated phytoplankton concentration is averaged over the upper 30 meters.

As shown by Fig. 19, G_OC_10km captures the observed spatial pattern of the yearly mean chlorophyll-a concentration, with low concentrations in the subtropical gyres and high concentrations in the equatorial region, North Atlantic, North Pacific and Southern Ocean (Fig.19 a,c). The concentrations are overestimated, but given the simplified representation of biology in

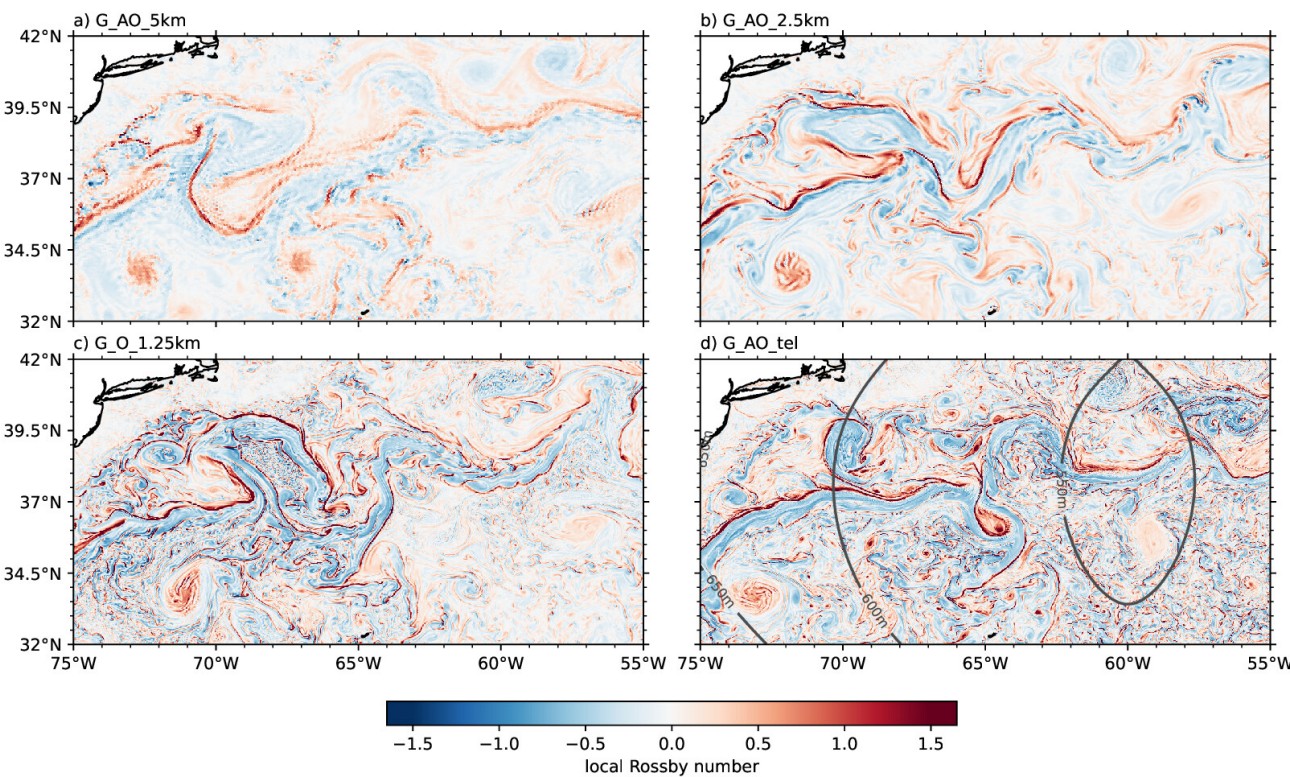

**Figure 18.** Local Rossby number defined as relative vorticity divided by planetary vorticity over the North Atlantic in the surface layer. The Rossby number is shown for (a) G_AO_5km, (b) G_AO_2.5km, (c) G_O_1.25km and (d) G_AO_tel. Snapshot taken after 40 days in (a-c) and 30 days in (d). In (d) contour lines depicting the grid spacing are also shown.

HAMOCC with the use of a bulk phytoplankton and the large uncertainties in the observations (35%), this first comparison looks generally promising. In the equatorial region, G_OC_10km overestimates the chlorophyll-a concentration due to nutri­ent trapping, similar to previous studies (Ilyina et al., 2013). This bias is reminiscent of the coarser 40-km results, used to initialize the biogeochemistry fields, and is expected to improve if the simulation would be run for longer. Moreover, zooming
into the North Atlantic region, Figs. 19c,d indeed reveal that G_OC_10km can capture the effect of mesoscale ocean eddies on ocean productivity, as expected from observations. G_OC_10km can also reproduce the seasonal cycle in chlorophyll-a concentration for the two hemispheres. In the Northern Hemisphere, the simulated seasonal cycle is in reasonable agreement with observations (not shown). It captures the spring bloom, but not the autumn one. In the Southern Hemisphere, the austral summer bloom is reproduced but with a much too strong amplitude, with simulated concentrations around 1.75 mg m$^{-3}$ in
December and January versus 0.25 mg m$^{-3}$ in observations. The overestimation is likely due to a lack of production in ice covered regions in G_OC_10km, leading to a large abundance of nutrients when the ice melts.



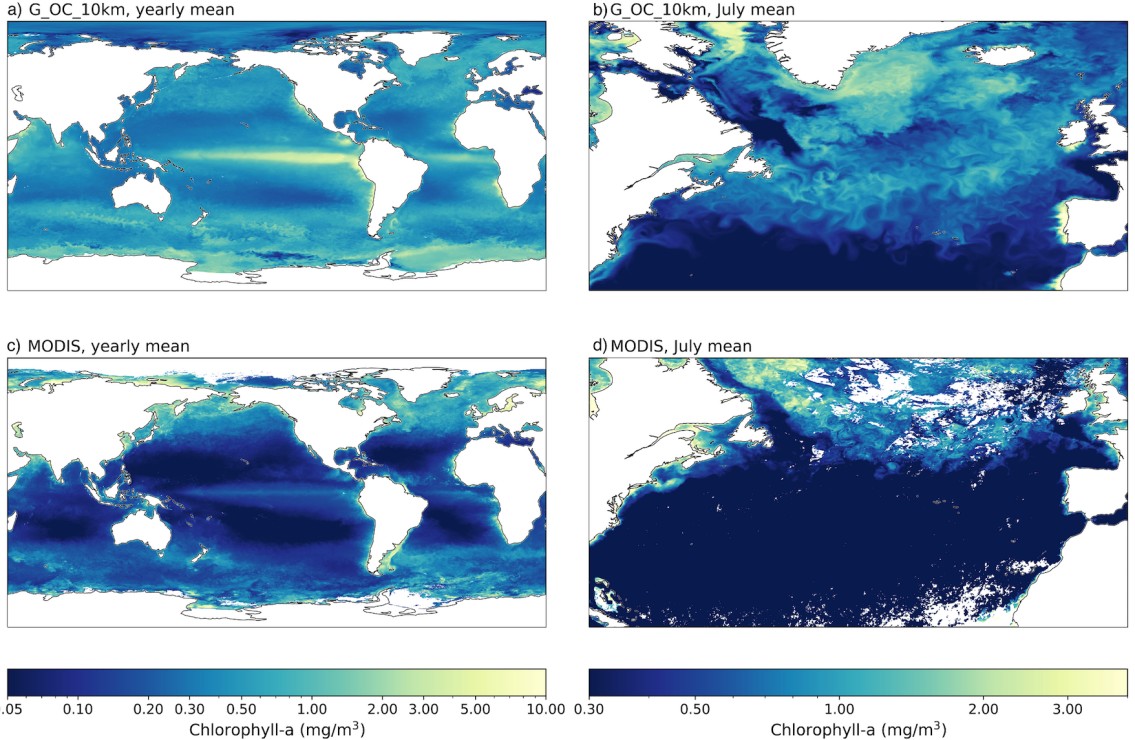

**Figure 19.** Chlorophyll-a from (a,b) G_OC_10km and (c,d) MODIS observations (grid spacing of 4 km) depicting (a,c) its global yearly mean distribution and (b,d) zoom into the North Atlantic for July 2016.

## 5   Conclusions

We have presented the new ICON-Sapphire model configuration developed at the Max Planck Institute for Meteorology in cooperation with the German Climate Computing Center, a configuration designed to resolve phenomena on small spatial scales

on the globe, with targeted grid spacings finer than 10 km. In contrast to models currently employed at such grid spacings, ICON-Sapphire contains all components of a traditional ESM, including an interactive ocean, ocean biogeochemistry and dynamical vegetation. We demonstrate that ICON-Sapphire can already now be run coupled for one year with a grid spacing of 5 km and for a few months with a grid spacing of 2.5 km. In the atmosphere, hectometer grid spacings can be achieved by using the ICON capability of inside nest and/or limited-area domain with prescribed or doubly periodic lateral boundary conditions.

In the ocean, hectometer grid spacings can be achieved by using a telescope feature over specific regions, both uncoupled and coupled to a uniform atmosphere. The highest achieved grid spacing in the ocean is 1.25 km globally and down to 530 m over the North Atlantic using the telescoping feature. Results of an ocean-only simulation including ocean biogeochemistry and integrated for 4 years with a grid spacing of 10 km are also presented towards demonstrating the ESM ability of ICON-Sapphire. All in all, compared to observations and expectations, we demonstrate that ICON-Sapphire performs satisfactorily

and is ready for a wide variety of Earth System applications and to investigate climate puzzles.





From the results of our 1-year global coupled simulation at 5 km with its sister simulation at 2.5 km for a few months, we find that ICON-Sapphire reproduces many features of the climate system, more or less out-of-the-box: the position and strength of the large-scale pressure systems and of the jet, the seasonal cycle in precipitation and soil moisture, salinity, the daily warming of the upper ocean in subsiding regions, the structure of the ITCZ in deep convective regions and the structure

of the atmosphere above the Gulf Stream in the mid-latitudes, as well as the coupling strength between SST, latent heat flux and precipitation over the ocean. The main shortcomings are a yearly negative radiative imbalance of 4 W m$^{-2}$ at the top-of-the-atmosphere, leading to a too low mean temperature, as well as a general overestimation of precipitation amounts except for a pronounced underestimation of precipitation at the equator. The equatorial region also stands out with too cold and too salty water. Finally, there are also some indications that the warming/cooling of the ocean surface does not propagate far

enough in the ocean interior. As ICON-Sapphire has not undergone extensive tuning, some of these aspects could be tuned away (Mauritsen et al., submitted), but they also highlight difficulties related to the representation of shallow convection in the atmosphere, as expected at such grid spacings, and mixing in the ocean. Moreover, the interactions between tropical SSTs and convection appear more sensitive than we expected. These challenges are defining new frontiers for the science and will have to be surmounted to realize the full potential of models such as ICON-Sapphire.

The employed grid spacing allows resolving interactions between small and large scales across components of the Earth's surface, as visually demonstrated by four examples: the development of a low-pressure system over the Southern Ocean with its associated imprint on the atmosphere and the ocean; the breaking of tropical instability waves with the development of secondary SST fronts and their imprint on the surface fluxes; the breakup of the Arctic sea-ice pack and the formation of leads and polynyas by a low-pressure system with its imprint on the surface fluxes; and the diurnal evolution of land-sea breezes

and their forcing on convective precipitation. Furthermore, we show that ICON-Sapphire can reproduce mesoscale patterns of trade-cumulus organization in a limited-area atmosphere-only simulation using a grid spacing of 620 and 308 m and that local Rossby numbers of order 1, indicative of submesoscale eddy activity, are produced with ocean grid spacings of 1.25 km and finer. Finally, from our 10-km ocean simulation including ocean biogeochemistry, we conclude that ICON-Sapphire can capture the effect of mesoscale ocean eddies on ocean productivity, as expected from observations.

The throughput of ICON-Sapphire with a grid spacing of 5 km is 126 simulated days per day using 600 nodes on the newest machine (Levante) of DKRZ, making integrations on multi-decadal time scales possible. Multi-decadal simulations at 2.5 and 1.25 km are still out of reach on a CPU machine like Levante. The atmosphere and land components of ICON have been ported and ran onto GPUs (Giorgetta et al., 2022), and a hybrid coupled ICON-Sapphire version using GPUs for the land and atmosphere and CPUs for the ocean is currently tested on JUWELS Booster. Extrapolating from past experience, we expect that

by using half of the LUMI machine, a throughput of 1 SYPD could be achieved at 2.5 km, a throughput generally thought to be needed for a climate simulation to be considered useful. As a next step, the ocean component of ICON-Sapphire will be ported to GPUs, as the performance of the ocean model becomes a bottleneck below a grid spacing of 5 km. Also, the atmosphere code is being simplified and refactored to better express the simplicity that arises when many of the parameterizations of traditional GCMs begin to be explicitly represented. One further aim of the refactoring is to better enable the scalable development to

different architectures. On the physics side, the use of thin layers in the upper ocean, possible with the implemented new





$z^*$-coordinate, the inclusion of river discharge, and a better representation of the net top-of-the-atmosphere radiation will be in focus. An important aspect of the conducted simulations remains workflow. Often data problems are data management problems, for instance orders of magnitude improvements in data access can be achieved by intelligently structuring and indexing the data. Moreover, encoding/decoding approaches from machine learning are being developed to interpolate from

sparse data, allowing to reduce disk space usage. Despite the remaining technical challenges, ICON-Sapphire already now allows performing unique multi-decadal climate simulations, with explicit interactions between small and large scales as well as between the components of the Earth System - ocean, atmosphere, land and cryosphere -, including carbon, at the forefront of exascale climate computing.

*Code and data availability.* Simulations were done with the ICON branch nextgems_cycle1_dpp0066 as commit62dbfc. This source code

is available here https://doi.org/10.17617/3.1XTSR6. The ICON model is available to individuals under licenses (https://mpimet.mpg.de/en/science/modeling-with-icon/code-availability). By downloading the ICON source code, the user accepts the licence agreement. The observational datasets were obtained from: https://www.cen.uni-hamburg.de/en/icdc/data.html except for ERA5, obtained from https://cds.climate.copernicus.eu/#!/home, for OAFlux, obtained from https://oaflux.whoi.edu/ and for PATMOS, obtained from https://observations.ipsl.fr/aeris/eurec4a/#/. The simulation outputs from the DYAMOND intercomparison project can be obtained from

https://www.esiwace.eu/services/dyamond-initiative/. Scripts employed to produce the figures can be found here: https://owncloud.gwdg.de/index.php/s/kn7GYFi3QmHmtFQ

*Author contributions.* All authors contributed to the development of the ICON-Sapphire configuration. Analysis of results was conducted by: C. Hohenegger, J. Bao, S. Bastin, M. Behravesh, M. Bergemann, R. Brokopf, N. Brüggemann, L. Casaroli, F. Chegini, G. Datseris, G. George, O. Gutjahr, H. Haak, J. Jungclaus, M. Kern, D. Klocke, T. Mauritsen, L. Paccini, D. S. Praturi, D. Putrasahan, F. Schütte, H. Segura,

R. Shevchenko, M. Specht, R. Vogel, C. Wengel, M. Winkler. The Sapphire project is lead by B. Stevens and C. Hohenegger. Manuscript was written by C. Hohenegger with inputs/comments/reviews from P. Korn, L. Linardakis, R. Redler, R. Schnur, J. Bao, S. Bastin, M. Behravesh, N Brüggemann, F. Chegini, G. Datseris, M. Giorgetta, O. Gutjahr, T. Ilyina, J. Jungclaus, D. Klocke, A. K. Naumann, L. Paccini, D. S. Praturi, D. Putrasahan, S. Rast, T. Riddick, F. Schütte, H. Segura, R. Shevchenko, C. C. Stephan, J.-S. von Storch, R. Vogel, M. Winkler, J. Marotzke and B. Stevens

*Competing interests.* No competing interests are present

*Acknowledgements.* F. Chegini, T. Ilyina, A.K Naumann and D. Putrasahan received funding from the Deutsche Forschungsgemeinschaft (DFG, German Research Foundation) under Germany's Excellence Strategy - EXC 2037 'CLICCS - Climate, Climatic Change, and Society' - Project number: 390683824. G. Datseris is funded by the European Horizon 2020 project CONSTRAIN, project number 493B. R. Vogel acknowledges funding by the European Research Council EUREC[4]A grant agreement 694768. S. Bastin, C. Hohenegger, J. Jungclaus, D.





S. Praturi and B. Stevens received funding from European Union's Horizon 2020 research and innovation program project NextGEMS under the grant agreement number 101003470. N. Brüggemann, O. Gutjahr, R. Shevchenko and J. Jungclaus received funding from the Collaborative Research Centre TRR 181 "Energy Transfers in Atmosphere and Ocean" funded by the Deutsche Forschungsgemeinschaft (DFG, German Research Foundation)- Project number 274762653. F. Schütte is funded by the Bundesministerium für Bildung und Forschung as part of the NextG-Climate Science-EUREC4A-OA project. T. Ilyina was supported by the European Union's Horizon 2020 research and

innovation program under grant agreement 101003536 (ESM2025–Earth System Models for the Future) and under grant agreement 820989 (COMFORT). T. Ilyina and F. Chegini were supported by the European Union's Horizon 2020 research and innovation program under grant agreement 773421–project "Nunataryuk". H. Segura is funded by the Hans-Ertel Centre for Weather Research under project number 4818DWDP1A. This research network of universities, research institutes, and the Deutscher Wetterdienst is funded by the Federal Ministry of Transport and Digital Infrastructure (BMVI). We acknowledge the mission scientists and associated personnel for the production of the

observational data, the modelling teams participating in the DYAMOND intercomparison project as well as DKRZ for use of its computer facilities.





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
