# Peer review of "ICON-Sapphire: simulating the components of the Earth System and their interactions at kilometer and subkilometer scales"

_Geoscientific Model Development, 2022_

## Referee Comment (RC2)

This paper describes an huge amount of cutting-edge work. It is well written and I have only minor comments. Basically, I think it is acceptable as-is. That said, I was a bit disappointed with the paper.

Major Comments:
1. Coupled km-scale simulations unlock a lot of interesting questions, but the results shown here seemed more of the "we made pretty pictures" variety. Having written overview papers for new model releases myself, I can commiserate – these papers are a lot of work to write and hard to make interesting to read. That said, I had a couple of questions going into the paper which might be useful to reflect on:
    a. **What features in the coupled system are improved by storm-resolving scales?** Some features can't help but become realistic as they become explicitly resolved. Orographic precipitation is an obvious example. Getting these things right should fix classic problems in coupled models like dynamic vegetation die-off in the Amazon due to precipitation biases or ocean circulation biases due to incorrect bathymetry. Identifying classic biases which you expect your GSRM to get right, then checking whether this happens would be interesting.
    b. **What features in the coupled system are NOT improved by storm-resolving scales**? Do you have a sense for what the canonical problems of GSRMs will be? What did you struggle to get right in your simulations?
    c. **What do we get out of global coupled k-scale models that was missing from prescribed-SST runs or regional simulations?** In this context, it would be nice to see prescribed-SST companion simulations and/or regional simulations.

2. I'm also disappointed by the discussion (particularly section 4.1.1) about:
    a. **Model tuning**: 4 W/m2 is a huge imbalance. I'm confused why you didn't insist on tuning the model better before running these simulations. How can you hope to do multi-decadal simulations with such a large radiative imbalance? I don't expect you to redo the simulations, but acknowledging the problem and explaining why you didn't want to or couldn't tune the model would be interesting. Ignoring the issue leaves the reader feeling like they missed something.
    b. **Model drift**: Fig 4 shows that TOA energy is adjusting rapidly over the course of this simulation. It would be nice to see a similar graphic for global-average surface temperature. Lack of discussion of what this drift means was conspicuously absent from the paper. Do you really think you could do a long simulation with this configuration? Do you have plans for fixing the drift?

3. I think a lot of the ocean analysis is naïve because it doesn't acknowledge that it takes the ocean a long time to drift away from its initial condition. Thus a lot of the analysis is probably more reflective of having an initial condition which looks like observations rather than that your ocean dynamics are working correctly. You can get a sense of initial condition versus equilibrated model bias by comparing the output from a coarse-resolution coupled run at initialization, after 1 year, and after 500 yrs. I bet most of your fields of interest look a lot more like the initial condition than the 500 yr value. I'm not sure this means you should

throw out your ocean analysis, but I do think you need to clearly articulate the potential source of good skill.

4. I'd like to hear more about conservation properties of the model. If I understand correctly, you've designed your schemes to have decent conservation properties and don't have a mass or energy fixer. I'd really like to see a plot of the unexplained global-average water and energy leak over time. This seems to me like it could be a huge problem for your mult-decadal simulation aspirations.

Grammar, spelling, and details:
1. ~L130: it seemed odd this paragraph doesn't include citations for readers to find out more about each of these component models, but then I realized that you go into a lot more detail about each component in following sections. If convenient, citations to the overview papers on each component model would be useful here. If not, please note that details about these models are given in section 2.
2. L135: You say here that the atmosphere can only be run in uniform global or regional modes, but Fig 2 and elsewhere in the text talks about nesting, which seems like it uses several different resolutions in one run. The text also seems to imply that nesting is only available in regional simulations, which seems odd. Why can't you do nested regions inside a global run? Also, I think of telescoping as identical to nesting: you divide each tile of a coarse outer grid into finer but uniform grid cells and run a regional version of the model on this patch of fine-resolution cells. This fits with the idea of a telescope extending in a few discrete segments rather than continuously deforming to extend and retract. I think the way your ocean model works is that resolution is allowed to vary smoothly throughout the domain.
3. Section 2.1: It would be good to mention in this section that aerosols are prescribed. You say this on line 303 in the I/O section, but readers will expect to hear about aerosol treatment in the atmos description.
4. Fig 3: Wow, this plot is cool – it has so much info. I'd prefer if dynamics and transport dots had their own line in the legend since closed and open dots obviously mean something uniquely different. Also, maybe add titles for the left and right columns of the legend since left is atm and right is ocean?
5. Is land seen as just another atm process? It seemed odd that some components coupled via YAC but land doesn't. A sentence explaining why would be useful.
6. I thought Bjorn said at Pan-GASS that you have another turbulence option (Deardorff?) which was unintentionally acting as a shallow convection scheme. Is that worth mentioning here in the same spirit as you mention 2 moment microphysics and RTE-RRTMGP but don't use it?
7. Does ICON include horizontal turbulent mixing, or just vertical? It seems like horizontal mixing will be important at the hectometer scales you run at.
8. L195: citation for Richtmyer and Morton numerical scheme?
9. L246 – unclear what "latter" refers to.
10. L278: What do "processes" refer to here? I think you mean the land model, atmosphere model, etc. I tend to think of these as "component models" with "processes" being

particular physics schemes within a component... but that might be idiosyncratic of me. The concept of "neighboring processes" seems odd since processes have no spatial relationship to each other. I think you mean the process called before or after in sequential time splitting?

11. L281: Doesn't the atmosphere just compute wind stress and provides that to whatever land model wants it? The way it's written, it sounds like the atmosphere provides a different wind stress to sea ice versus ocean.
12. L286: I think you should delete "and" between "wind" and "vectors"
13. L298: It would be handy to point out that 30" is equal to ~900m at the equator.
14. L325: "single-precision 32-bit float arrays are now kept in memory". I think you mean that "output is now stored in single instead of double precision, reducing memory requirements by a factor of 2". "kept in memory" sounds like you mean the data is kept in cache instead of slower-access disk and "single-precision 3d bit" is redundant.
15. L175 – I'm confused how you obtain good performance on GPUs if you use PSrad for all calculations in this paper and PSrad only runs on CPUs – I would have thought having some processes on CPU and others on GPU would result in excessive communication overhead and slow runs. Is radiation running in parallel with other atm processes? Or is it just that you call radiation so infrequently (every 15 min!) that it doesn't matter? I suspect you are forced to run radiation so infrequently precisely because it is on CPU.
16. L339: I think 40 TB/month is for *5 km dx*. It would be useful to point this out and to also say how much storage space you're using for the 2.5 km grid (which I expect is 4x more = 120TB/month!).
17. L369: "half productive" is bad grammar.
18. Adding another panel to Fig 4 showing the annual cycle of global-average surface T would be useful.
19. L447: observations misspelled.
20. L452: I'm unclear how negative TOA radiative imbalance would lead to enhanced radiative cooling. First, what's your sign convention? Does negative radiative imbalance imply that the planet is losing heat? If so, I would think enhanced radiative cooling would *cause* the radiative imbalance. But TOA radiative imbalance could also be caused by an excessive planetary albedo.
21. Fig 11: caption skips panels e and g.
22. Fig 12: panel a and e seem redundant.
23. L554: "my" should be "may"
24. Fig 15: I thought sea breeze was a weak example of 2.5 km resolution since it is also captured pretty well at 25 km resolution. Also, this graphic would be a lot better using wind vectors rather than colors for just zonal wind.
25. L586: "latter" rather than "later"?

---

## Author Comment (AC1)

We thank the first reviewer for his thorough reading of our manuscript and insightful comments that helped clarify our manuscript and strengthen our validation. For simplicity, we rewrote the reviewer's comment below (in black) and respond to them point-by-point (in blue).

————————————————————————————————————————

**General comments**

In this article the authors describe a new model ICON-Sapphire that has been run at a variety of very high global resolutions of 10km and finer for coupled and ESM simulations, in order to explicitly represent processes that have previously been parameterised. Model simulations are short, one or multiple years for some of the reference simulations. Some initial evaluation of the model simulations is shown as well and information on model performance is given.

The article is well aligned with GMD in terms of model development and description, and very useful to inform the community of such impressive progress in terms of global model resolution advancements and future prospects for their use in climate simulation.

Thanks.

My main concern is the model evaluation part of the manuscript. I understand that it is difficult, with only one year of coupled simulation (which is not initialised and hence difficult to compare to any particular observational data year), to produce evaluation of the model. However, I have significant concerns about the evaluation shown. In particular with regards to the ocean, it is terribly unclear how the one year of simulation differs from its previous spin-up, and hence what the impact of coupling ~5km atmosphere and ocean models together is in that one year. Perhaps by showing similar diagnostics at the start and end of the one year simulation, or some differences over that one year period, it would be clearer what the simulation achieves.

We understand the reviewer main concern about the model evaluation part related to the ocean very well where, with a one-year simulation, the slow ocean dynamics can a priori not drift too far away from its initial conditions, an issue that we didn't explicitly discuss in the submitted version. The reason we think it is still important to validate the ocean state, even for one year, is that one could already investigate interesting scientific questions just based on one year of simulation, and we thus have to make sure that there is no obvious bias, even in the ocean. Despite the long experience of running ICON at low resolution in a coupled mode (Jungclaus et al., 2021), that the ocean works and couples correctly to the atmosphere is not given. In fact, several bugs were actually discovered during the development phase, bugs related to the momentum coupling between ocean and atmosphere. We thus added in this revised version both the note of caution related to the slow

ocean dynamics and the motivation for still validating the ocean at the beginning of section 4.1 on lines 423-431.

Second, based on the suggestions of both reviewers (see below for our detailed answers to the specific comments), we analysed how much of the biases we saw in G_AO_5km are just inherited from the spin-up, or are developing through the simulation. We conducted this analysis for salinity (Fig. 7) and for the barotropic streamfunction and water transport (Fig. 9), see our response to the corresponding reviewer comments below.

I suspect the above problems are enhanced by the imbalance in the top of atmosphere radiation. I understand that this is a "tuning" aspect that has not yet been addressed, but a more explicit showing of this bias (and presumably the inherent surface temperature and other consequent biases) would give a clearer view of the current capability of the model rather than what the model might be able to do in future.

The reason we originally kept this part rather short is that there is another paper by Mauritsen et al. 2022 (now published) that specifically deals with this aspect of tuning. But from the comments of both reviewers we now feel that we should say more on this aspect. First, we added the seasonal cycle of surface temperature in Figure 4. Second, we explicitly explained our current strategy to get rid of this imbalance. The TOA imbalance is mostly related to a preponderance of low clouds. These arise when using the Smagorinsky scheme, which has a mixing cutoff. Our formulation sets the eddy diffusivities to zero if the Richardson number is greater than the eddy Prandtl number. This cutoff unrealistically inhibits mixing, both because of well known limitations of the Smagorinsky scheme in simulating the transition to turbulence (Porté-Agel et al., 2000), and because of a failure to incorporate the effect of moist processes. As a result, over cold and moist surfaces, insufficient ventilation of the boundary layer occurs, causing moisture to build up and resulting in excessive low clouds. In ongoing experiments, we have explored adding a small amount of background mixing at interfaces between saturated and unsaturated layers where the equivalent potential temperature decreases upward, mimicking the effects of buoyancy reversal (Mellado, 2017). Low clouds respond sensitively to this background mixing, what provides a convenient control on their amount and on their influence on the top of the atmosphere energy budget. Ongoing work is exploring theoretical justifications for the choice of the background mixing, but it may also be set empirically, as a way to provide a better representation of the statistics of low clouds. We added these considerations on lines 450-461. Finally we partly rewrote the second paragraph of the conclusions to better summarize the current capability of ICON-Sapphire (see lines 715-726). Many investigations do not require long simulations, e.g. investigating processes that control the seasonal migration of the rainbelts, or interactions between ocean-atmosphere and ice as shown in Fig. 14, or effects of small-scale ocean features on large-scale properties of the atmosphere. But it is clear that the current imbalance prevents decadal simulation as the model cools

too much. So although the simulations may not be adequate to answer every question yet, we find it important to make the community aware of those simulations as they already allow to investigate aspects of the climate system in a novel light.

Several of the comments below refer to phrases used such as resolving convection/convective storms. I will not insist, but my colleagues more expert in convection processes than me would point out convection happens all the way to the sub- metre scale, so perhaps you could make this point early on in the manuscript and then use whatever phrase you choose.

Yes we agree. There are different terminologies in use in the literature and different opinions on it. We will use "represent convection explicitly" when talking about convection specifically as, we agree, we don't resolve the full spectrum of convection, and "resolve storms" as, in terms of storms, we do resolve them. Resolving storms also corresponds to the new terminology in use for atmosphere-only kilometer-scale models which are called (by a part of the community) "storm-resolving".

**Detailed comments**

L27: "grid spacing of 10 km would at least permit..." – I think I understand the sense of this, but maybe it could be more explicit. I think you are saying 10 km is enough to use explicit representation of convection (i.e. switch off the parameterisation of convection), just saying permit is a bit imprecise on this point.

Yes we meant that 10 km is enough to use an explicit representation of convection. We rephrased accordingly, see lines 28-29.

L57: Again possibly semantic, but "resolved explicitly" perhaps could be "represented explicitly".

Changed accordingly (line 58).

L83: again, *resolving* convective storms vs representing them.

For storms (see above), we will keep resolve but we will remove explicitly as it sounds a bit redundant (lines 83-84).

Fig. 3: I confess I struggle to follow the logic of the timestepping illustrated with so many lines and symbols.

We are in sympathy with the reviewer comment but we haven't found a better representation. Given the technical flavor of GMD, we find it important to show such a figure. Also the figure conveys very well the complex and maybe questionable logic of the timestepping in ICON.

L225: If I understand, the rainfall over land is not put back into the ocean currently? Given that you talk about the importance of water and energy cycles, is this not a big problem (perhaps it would be once the model is run for longer)?

We agree that this would be a big problem for longer simulations. The reason we turned the discharge off was that all the discharge of one river was happening in one ocean grid cell, which is an incorrect assumption with a grid spacing of 5 km. Moreover, the river reservoirs need to be properly initialized, otherwise too much fresh water ends up in the ocean in the first few months of the simulation. These two aspects have been since then corrected and new ICON-Sapphire simulations can use discharge. We adapted the text to clarify this, see lines 236-238.

L 245: This text suggests (but does not explicitly say) that Z* is used – perhaps it could be clearer if that is the case?

No z is used, we added this information on lines 256-257.

L260: I guess more details may be in Korn et al, but perhaps a little more detail would be useful on the sea-ice. Is EVP not marginal at these scales in terms of its assumptions? Perhaps this could be mentioned, or else justified, given similar discussion of the atmosphere setup. Also a single category zero-layer thermodynamics model is relatively simple these days and could be noted?

We agree with the reviewer's intuition that with kilometre and subkilometre scales we approach regimes where some of the standard assumptions of sea-ice modelling may become questionable (see e.g. Rigeisen at al, https://doi.org/10.5194/tc-13-1167-2019). The reason for choosing the EVP rheology in this work was its superior computational efficiency compared to the VP rheology (and this was the primary motivation for developing EVP at all). The question when VP or EVP starts to become invalid is hard to answer. Recent work (Koldunov et al, https://doi.org/10.1029/2018MS001485, Spreen et al https://doi.org/10.5194/tc-11-1553-2017, Wang et al https://doi.org/10.1002/2016GL068696) suggests that the EVP rheology provides at high-resolution of 4.5 km a good compromise between physical realism and computational efficiency. To take the reviewers comment into account, we have changed the corresponding part of the text, see lines 270-273.

L290: Much emphasis at the start of the paper is the importance of moisture and energy. But it is unclear how well the coupler conserves fluxes of heat and moisture across the interface. You say no correction is applied for global conservation, but do you know how well the model conserves globally and can this be stated?

Most configurations use the 1-nearest-neighbor exchange of data, as the grid geometry is identical between atmosphere and ocean over the area of exchange. Thus, conservation of quantities by the interpolation is not an issue here.

Conservative remapping is only applied in G_AO_tel. We did not investigate global conservation properties for this particular model configuration. From century long simulations of the coarse resolution model version described in Jungclaus et al (2021), we know that e.g. the water in this model configuration is closed (which includes conservative remapping of evaporation and precipitation). To clarify this, we rephrased the sentence "No correction is applied for global conservation" to "The coupler and our coupling strategy are designed to conserve fluxes", see lines 301-302.

L305: Does the land surface not need some spin-up too, given the different resolutions of the initial data and the model?

We didn't spin up soil moisture as it is not clear how to best do this. We cannot run the model over many years to wait until soil moisture equilibrates and starts from this state, as this is computationally too expansive. The other possibility would be to run the land model offline, either forced by observations or by coarse-resolution model output. But in this case, the land surface model would equilibrate to the observation climatology or coarse-resolution climatology, which does not have to be the same as the one obtained if forced by the 5-km atmosphere simulation. We checked the evolution of soil moisture (see Fig. R1 below). Despite the lack of spin-up, four out of the five soil layers are already equilibrated by September in the northern mid-latitudes, and three out of the five layers in the tropics. We added this information in the text on lines 318-321.

L323: Perhaps this bit could go into an appendix, rather interrupting the flow about the model

We agree that this part might interrupt a bit the flow. We shortened it (see lines 340-341), avoiding the description of the CDO operators, a description which is now available in the new release of the CDO documentation (https://doi.org/10.5281/zenodo.7112925).

L336: Maybe these numbers could go in a table, they might then be easier to digest?

We agree and did a table, now Table 2 and correspondingly adapted the text on line344-345.

L357: Similarly there are lots of numbers in this paragraph too, making it hard to absorb.

We simplified this paragraph by only talking about numbers in PFlops (rather than PFlops and nodes) and removed unnecessary repetition of the grid spacing, see lines 368-380.

Table 2: Just to check, for Δz: L, is the vertical spacing over 5 levels giving 5700m on the last level?

5700 m is the thickness of the last layer, 0.065m the thickness of the first layer. This is different in the ocean, where the first layer is thicker than the second layer. We clarified this in the caption of Table 2 (now Table 3).

L395: Ocean spin-up. I'm not sure I understand this, and I don't understand the metric of the biases at L402 (global mean?). The biases in a forced ocean model are generally rather constrained by the forcing (certainly at the surface). A figure would help to illustrate what the biases look like spatially, since global mean biases can hide compensating errors.

We rewrote the beginning of this part to better explain how we spin up the ocean, see lines 406-409. Concerning the bias, we agree and rephrased the corresponding lines (see lines 416-420). As shown in Fig. R2 below, the SST is consistently too cold except along 60S, north of 50N and along the western coast of North America. The global mean SST bias is -0.53 K. Since the SST bias does not show too much spatial structure, and since we already have many figures, we didn't include a figure but indicated in the text where the bias is positive/negative. We also do not show a figure for the salinity bias (see Fig. R3b) as the bias tends to be small due to the use of salinity relaxation, except in the Arctic. This is now mentioned on lines 419-420. See also our reply to the comment related to Fig. 7.

In addition, can you clarify that the spun-up ocean state uses all prognostics variables – temperature, salinity, currents, sea level height etc. It is unclear, later on, when you compare the simulation with observation, how much is included in the initial state.

We confirm that the spin-up ocean simulation employs all prognostic variables, an information that we added on line 426.

Before 4.1.1: perhaps you could say something in this section about the difficulties of assessing models on such short simulations, given issues with spin-up, coupled model shock, lack of TOA balance, lack of directly comparable observation year etc. I think this echoes through the next sections, but is being slightly ignored and I'm sure this is an issue that future groups will grapple with too. As mentioned in the overall comments, a more convincing assessment of the one year of coupled model data (that is demonstrably different from the ocean spin-up, for example) would be very welcome.

We agree. We added before the beginning of 4.1.1 (lines 421-431) the difficulty of comparing to observations due to the lack of directly comparable observation year, which motivated our use of year-to-year variability in the observations besides the climatological mean in our previous analysis. We also now mention both the difficulty of validating the ocean given the shortness of the integration period as well as the motivation for still doing so. We don't mention the lack of TOA balance yet, as this is a result of the validation. And as described below in response to the specific comments concerning Fig. 7 and Fig. 9, we added a more convincing assessment of the one year of coupled model data versus its spin-up.

To a suspicious eye, it also feels a little that metrics have been chosen where they look good, and others mentioned but any poor representation is blamed on lack of TOA balance or similar. Could you perhaps better motivate the metrics you show to convince readers otherwise?

We disagree with the reviewer's comment. We looked at obvious basic features of the climate system, namely TOA radiation balance, temperature, large-scale circulation of the atmosphere, precipitation, soil moisture, salinity, ocean circulation and ocean-atmosphere interactions. We believe that our comparison was fair, but what may have led to this impression is that, in the more summarizing sentences of the manuscript, we generally only stated that ICON-Sapphire reproduces basic features of the climate system. We now added in the abstract (see lines 9-10) as well as at the beginning of section 4 (see lines 393-394) that ICON-Sapphire can reproduce basic features of the climate system even though some aspects would require further improvements. We also removed the last summarizing paragraph of section 4.1.1 which only summarized the positive aspects, and partly rewrote the second paragraph of the conclusions to better highlight what we can capture well and what remain bottle necks (see lines 715-726).

L427: but also that you have not spun-up the land surface, it is just taken from initial conditions? Presumably one can expect the land surface to take considerable time to spin- up, e.g. soil moisture etc.

The soil moisture indeed needs time to spin up, but surprisingly, this seems to happen quite fast, see our reply to above comment related to L.305 and Fig. R1 below.

L428: it is a shame that spatial plots are not shown here but referenced to another paper that is not available yet.

We apologize. We didn't want to include the plots as we already have many plots in the paper and those plots are indeed in another paper. That paper has now been published (https://a.tellusjournals.se/articles/10.16993/tellusa.54/). Note also that, in response to the comments about the TOA imbalance and its effects, we now added a new panel in Fig. 4 showing the seasonal cycle of temperature.

L429/430: Can you include a reference for mid-latitude storms dominating meridional transport, convection for vertical transport?

We rephrased the sentence to "In the atmosphere, storms (baroclinic eddies) dominate the  energy transport in the extratropics, whereas atmospheric convection dominates the vertical energy transport in the tropics.", see lines 475-476.

Fig 5: Based on 1 year (1 season) of data. Is this robust – e.g. different runs, or longer runs (I know these have not been done)? Any climate analysis would consider 1 year to be far too little data to base assessment on, also given the TOA.

Are you saying that the TOA makes no difference to the large scale circulation? How much is the simulation able to change from the initial conditions in such a short time?

We indeed only have one year for the simulation, but for the observations, we included 10 individual years to get an estimate of internal variability. We agree that, although the first year falls into the internal variability, we cannot exclude that the second year might not, but a priori, we would not expect this to be the case. The atmosphere has a very short memory, weather systems loose predictability in about two weeks, so if there was something really wrong, we should have seen it already in the first year. Also weather forecasts, whose main aim is to predict the large-scale circulation of the atmosphere, are conducted with models that have TOA imbalances. Maybe if we would run the model for many years and maybe if the TOA imbalance would change the temperature gradient drastically, this could have an effect on the large-scale circulation, but we don't want to go into too many conjectures here, also because we will fix the TOA imbalance before running decadal simulations.

Fig. 6: My looking at the "all points" figure would also suggest a double ITCZ through the year in the model, with ITCZ not fully propagating across the equator.

Yes, the issue is that G_AO_5km keeps two clear bands of rain over the western Pacific at all times. As mentioned in the text, the equator stands out as very dry (in terms of precipitation) and this leads to two clear bands of precipitation with the southern band being too zonal, reaching too far east and remaining south of the equator even during boreal summer. We added this information in the text when discussing Fig. 6 on lines 490-492.

L460: As noted before, and perhaps it is a small point, but convection happens at all scales. You are resolving the part above your grid spacing, but I think many would argue you are not resolving convection, you are explicitly representing it. As you suggest above with shallow convection and clouds, you are still missing some important processes, which you are choosing not to parameterise.

We agree and changed the text to "explicitly representing" as noted above (see line 506).

L465: You state here that the hydrological budget is closed in ICON-Sapphire, but you have not shown that previously. Can you say more about this earlier in the manuscript? Indeed it is not closed in the usual sense, as you do not have the rivers returning fresh water to the ocean, as I understand it.

We meant the hydrological budget over land, which we checked and was closed. We added "over land" to clarify (line 512).

L466: The low soil moisture, is this the initialisation or the simulation?

As shown from Fig. R1 below, it is not from the initialisation but from the simulation as soil moisture decreases with time. We added this information (line 513).

L468 & Fig. 7: The spin-up of the ocean is presumably forced by observed precipitation, and (somehow, undefined) I assume the ocean salinity is constrained during spin-up? If so then this figure is a given (is it not, if not then please say why), because in one year the model will not change these large-scale patterns and hence you are mostly showing the spun-up state.

Yes, as usual in uncoupled ocean simulations there is sea surface salinity relaxation active. The salinity is relaxed towards observed 10m salinity of the PHC3 climatology with a time constant of 3 months. We added this information on lines 413-414.

We partly disagree with the reviewer that we are mostly showing the spin-up state. As discussed previously in the paper, salinity biases arise at the mouth of big rivers, because we neglect river discharge, and, salinity biases in the tropics, are consistent with precipitation biases. Hence, being related to the absence of river discharge and to the simulated precipitation pattern, the pattern of the salinity bias in the tropics is distinct from the one at the end of the spin-up period. This is confirmed by Fig. R3 below where we show the salinity bias in the last full month of the spin-up (December 2019) and the same month in G_AO_5km (December 2020). Except for the salinity bias in the Arctic, the pattern looks distinct. To clarify this point, we updated the discussion of Fig. 7 in the text, now mentioning which biases are inherited from the spin-up and which not, see lines 517 and 521-523..

Figure 8: I don't understand what the zonal mean of the correlation (which is not clearly defined in the text) is meant to show. Why not show spatial maps (as in Wu et al) to demonstrate where the correlations are positive/negative and hence suggest mechanisms.

For each grid point, we computed the correlation between (a) SST and latent heat flux and (b) SST and precipitation and then took the zonal average of that value. We clarified this in the text on lines 524-525. The reasons for showing zonal means rather than spatial maps are twofolds. First, this allows us to indicate the internal variability in the observations. As mentioned by the reviewer in one of his previous comments, validating one year of simulation is difficult as it is not clear which year the model is representing. Second, we were struck by Fig. 3 of Wu et al., where the correlation between SST and precipitation was positive everywhere in the climate model whereas there were clear meridional differences in observations. To check this, a cross-section of the zonally averaged correlation coefficient is sufficient. We added this motivation for looking at the zonally averaged correlation in the text, see lines 525-528. Understanding differences in the spatial pattern of the correlation would be topic for an own study, as shown by the paper of Wu et al. dedicated to that question.

Figure 9: Can you add the uncertainties for the observations into the table. At the moment it implies much better known values (in 2020 or any other year) that is actually the case, and add to the caption that the observed values are for (historic) periods, not 2020.

We added the uncertainties for the observations into the table, as given in Griffies et al. (2016) and in Donohue et al. (2016) for the Drake passage. We also added to the caption that the observed values are for historic periods.

L484: Again I have some trouble with this. If the ocean was spun-up with observed forcing, then of course after 1 year it will still look like that state. Can you say anything about what changes in the ocean over the 1 year that would suggest otherwise?

Following the reviewer's suggestion, we compared the transport values obtained in one year in G_AO_5km to the transport values from the spin-up simulation, see the updated Table in Fig. 9. Compared to the mean transport obtained in the spin-up, the transport, with the exception of the Bering Strait, is weaker, with values out of one standard deviation (computed to quantify internal variability) for all the passages but the Indonesian Throughflow and the Mozambique Channel. Hence the coupling leads to systematic differences. Except for the Florida Bahamas Strait, the weaker transport of G_AO_5km is in better agreement with observations. The weaker transport is consistent with weaker wind stress in G_AO_5km compared to the spinup simulation, also expressed in a weaker barotropic streamfunction. We added these considerations on lines 540-544.

L495: "..which may affect the Indian summer monsoon" – do you have a reference for this statement?

We added a reference to Seo (2017), see line 550.

L497: I think I might be able to see one TC path in the North Atlantic in JJA, but they are not obviously visible to me otherwise unless you label them.

We didn't label TCs as this would imply that we indeed track them. We thus reformulated the sentence to "Tropical cyclone tracks are also visible, for instance off the coast of central America in the eastern Pacific in JJA . " to highlight one particular example (see lines 552-553).

Figure 11: Please label the solid and dashed lines in panels e&g. I'm also not quite sure that the d&f panels are meant to convey, without any reference to observations/reanalysis to compare against.

We added labels. Panels d&f have only illustrative purpose. Since this kind of simulations are new, and structurally very different from low-resolution models, which we now recall when discussing Fig. 11 on lines 555-556, we find it interesting to illustrate how the ITCZ and the atmosphere over the Gulf Stream look

like in such simulations. Also observations from vertical velocity (in d&f panels) would have to be taken from field experiments, but we don't have a field experiment coinciding with the simulated period. This is different in panels a-c where we could make use of the EUREC⁴A field experiment which happened right at the start of the simulation, where one can expect the simulation to still reproduce the observed weather.

Figure 13: I assume that in (b) the region shown is blown up in size, if so please can you add that to the caption.

We added this to the caption.

Figure 15: I'm struggling to take much from this figure. The precipitation is very noisy in the contours and it is difficult to see the interaction between wind and precipitation as suggested.

We removed this figure as we agree that it is not the best representation and, as also pointed by the second reviewer, there is no surprise that the model can reproduce land-sea breezes. We just kept this information in the text, but also shortened that paragraph, see lines 600-601.

L547: Do you mean minimalistic parameterised physics?

Yes, but since we removed that paragraph, comment doesn't apply anymore.

L572: Could you plot the 1.25 km driving model data as well. As with the figures above, it is difficult to see that you are demonstrating this model can do something different from other (e.g. lower resolution) models when you are only presenting one figure and no process validation.

We are not sure that showing the 1.25 km driving model will add much information. With a grid spacing of 1.25 km, the driving model is also able to reproduce the observed features. A model with a much lower resolution would not be able to do so, but we don't have such a simulation. Also the goal of this section is to show that we can not only run ICON-Sapphire globally but also on limited areas, which is of interest for many applications. But we on purpose don't do an extensive validation as there is nothing very novel in being able to reproduce patterns of mesoscale organization with high enough resolution on limited domains and we wanted to concentrate in the paper on the more novel aspects. We adapted the beginning of section 4.2 (lines 609-610) to better convey this message and not raise false expectations. Having said this, note that we do validate the simulated cloud cover on lines 629-633.

L623: As before I'm not convinced that Fig. 19 says very much about the performance of the 10km model. You note that the bias is inherited from the spinup, so I think it would be useful to show the spinup field as well, and/or the

difference to the 10km model. You do not show how much this field is reset each seasonal cycle, hence it is unclear what the 10km model is contributing.

The most important contribution of the 10km model is that it captures the mesoscale eddies in contrast to the coarser resolution 40km model. The imprint of these eddies in the 10km model are seen, for example, in the phytoplankton field in the North Atlantic in July (Fig. R4), whereas the fields simulated by the 40km model are much smoother. Previous studies have shown that the impact of mesoscale eddies on local carbon export production can be as large as 50% (Harrison et al., 2018). We agree that in terms of the large-scale pattern, differences between the 10km and the 40km simulations are small, but we can already see differences in the seasonal cycle (Fig. R5) where the performance of the 10-km simulation is slightly better than the one of the 40-km.

Taking these considerations into account, we added the following paragraph at the end of section 4.4 (lines 689-696): "The large-scale pattern displayed by G_OC_10km in Fig. 18a is reminiscent of the large-scale pattern displayed by the 40-km spin-up simulation. The two fields correlate with a correlation coefficient of 0.89 for the yearly mean. However, the mesoscale structures displayed by Fig. 18b are clearly absent in the spin-up simulation and we can already see differences during the blooming season, with G_OC_10km simulating lower chlorophyll concentration than the spin-up simulation, in better agreement with observations. In the northern hemisphere, peak values are 1.5 mg m$^{-3}$ in the 40-km spin-up simulation, 1.25 mg m$^{-3}$ in G_OC_10km and 0.88 mg m$^{-3}$ in observations. Whether these differences are due to the representation of mesoscale eddies is too early to tell, but is a further argument for being able to run ICON-Sapphire as an ESM on longer time scales."

**Technical corrections**

L103: ...climate processes *are* represented physically – perhaps *explicitly* rather than physically?

Changed accordingly (line 105).

L366: Finland

Corrected (line 377).

L383: "already now" – perhaps just one or the other, they both mean the same thing.

Corrected (line 396), also other instances in the text.

L447: observations

Corrected (line 494).

L588: simulation

Corrected (line 641).

[Figure]

Figure R1: Time evolution of soil moisture in each soil layer and averaged over 30N-60N (left) as well as 30S-30N (right).

[Figure]

Figure R2: SST bias (K) computed as the monthly mean difference between the spin-up simulation and the Ocean Reanalysis System 5 for the last full month of the spin-up period (December 2019).

a) G_AO_5km

b) Spin-up

[Figure]

Fig. R3: Monthly mean (December) salinity bias (g kg⁻¹) in G_AO_5km and in the spin-up ocean simulation. Observations from PHC climatology.

[Figure]

Fig. R4: Chlorophyll-a concentration in the 10km and 40km simulation over North Atlantic in July.

[Figure]

Fig R5. Seasonal cycle of chlorophyll-a concentration in the Northern (left) and Southern (right) hemisphere simulated by the 40km and 10km model, compared to observations.

---

## Author Comment (AC2)

We thank the second reviewer for his thorough reading of our manuscript and insightful comments that helped clarify our manuscript and strengthen our validation. For simplicity, we rewrote the reviewer's comment below (in black) and respond to them point-by-point (in blue).

————————————————————————————————————————

This paper describes an huge amount of cutting-edge work. It is well written and I have only minor comments. Basically, I think it is acceptable as-is. That said, I was a bit disappointed with the paper.

Thanks. We understand the reviewer's comments and feeling, but we want to reiterate here that this is a model development paper to present our new ICON-Sapphire version. The goals were to document the model code, to show that km-scale global coupled simulations are technically feasible and to show to which extent basic features of the climate system can be reproduced by our current set-up (see lines 102-106 in the introduction). The reviewer raises many interesting scientific questions in his major comments, like the effect of small scales on larger scales, or the impact of the coupling, questions that our new ICON-Sapphire version would indeed allow to answer in the future. But before such questions can be answered, we need to have a reference paper that describes the model and its current capability, upon which future (and more interesting studies) can build upon.

Major Comments:
1. Coupled km-scale simulations unlock a lot of interesting questions, but the results shown here seemed more of the "we made pretty pictures" variety. Having written overview papers for new model releases myself, I can commiserate – these papers are a lot of work to write and hard to make interesting to read. That said, I had a couple of questions going into the paper which might be useful to reflect on:
As replied below in detail, we reflected upon the questions raised by the reviewer. Although we showed a couple of "pretty pictures" (current Figs. 12, 13, 16 out of the eighteen figures), we believe we also presented a comprehensive validation of basic features of the climate system, even considering internal variability in observations to better assess the skill of the presented one-year simulation, where the short integration period complicates the validation.

a. What features in the coupled system are improved by storm-resolving scales? Some features can't help but become realistic as they become explicitly resolved. Orographic precipitation is an obvious example. Getting these things right should fix classic problems in coupled models like dynamic vegetation die-off in the Amazon due to precipitation biases or ocean circulation biases due to incorrect bathymetry. Identifying classic biases which you expect your GSRM to get right, then checking whether this happens would be interesting.
As this is the first paper presenting our ICON-Sapphire version, and since there was already a lot of material to cover (description of the code, presentation of

different set-ups), we decided to concentrate on an evaluation of the representation of basic features of the climate system against observations. In that context, we presented a comprehensive validation against observations, checking basic climate variables and properties of the climate system, including internal variability in observations (in current Figs. 4 to 9, 11, 15, 18). Adding a comparison to low-resolution simulations to assess which features of the coupled system are improved by storm-resolving simulations would be one obvious next step, we agree, but this exceeds the frame of the current paper  and is left for future work.

b. What features in the coupled system are NOT improved by storm-resolving scales? Do you have a sense for what the canonical problems of GSRMs will be? What did you struggle to get right in your simulations?
One delicate issue is to get the TOA energy budget right. First, some of the known tuning knobs from low-resolution climate models don't exist anymore and from those which still exist, the simulations often didn't react as expected. Second, trade wind cumuli play an important role in determining the TOA energy budget and getting them right can be challenging, especially in our case as we are trying to avoid having to use a shallow convective parameterization. The second delicate issue may be the coupling between the atmosphere and the ocean. The atmosphere turned out to be (at least to us) surprisingly sensitive to small difference in initial SSTs. Throughout the development phase, we had instances where the winds, e.g. in the tropical Atlantic, switched from easterlies to westerlies. This may also explain the difficulties of our simulation to capture precipitation at the equator. The last delicate issue seems ocean mixing, e.g. we have too shallow ocean mixed layers in the tropical Atlantic (see Fig. 11) and simply tuning the mixing coefficient in the TKE scheme didn't fully alleviate that problem. In this revised version, we first expanded the discussion about the TOA imbalance (see lines 450-461). Second, we also added some details concerning the issue with ocean mixing on lines 563-565. Third, we more explicitly mentioned in the goals (see lines 105-106) that our validation can shed light onto potential remaining shortcomings of GSRMs and accordingly partly rewrote the second paragraph of the conclusions (see lines 715-723) to more explicitly mention what we are not getting right and what the canonical problems of GSRMs might be given our experience with the development of ICON-Sapphire. Finally, we also added both in the abstract (see lines 9-10) as well as at the beginning of section 4 (see lines 393-394) that, although several features of the climate system are well captured, not every feature is well captured. We agree that our previous formulation was too positive as, as already described in the previous version, not every aspect of the climate system is well captured.

c. What do we get out of global coupled k-scale models that was missing from prescribed-SST runs or regional simulations? In this context, it would be nice to see prescribed-SST companion simulations and/or regional simulations.
This is a very important question, we agree with the reviewer, but it would justify a paper on its own.

2. I'm also disappointed by the discussion (particularly section 4.1.1) about:
a. Model tuning: 4 W/m2 is a huge imbalance. I'm confused why you didn't insist on tuning the model better before running these simulations. How can you hope to do multi-decadal simulations with such a large radiative imbalance? I don't expect you to redo the simulations, but acknowledging the problem and explaining why you didn't want to or couldn't tune the model would be interesting. Ignoring the issue leaves the reader feeling like they missed something.

We agree with the reviewer that the imbalance prevents performing long simulation as the model cools too much with time. The reasons for not insisting on tuning the model are twofolds. First we wanted to know what we get right more or less out of the box, just by trying to represent explicitly as much as we can from the climate system. Second, many scientific questions, e.g. dealing with the coupling between convection and SST, or effects of small-scale oceanic features on the large-scale circulation of the atmosphere, can be already tackled with a one-year simulation, and for the first time without being concerned that many of these features are the result of the design of a convective parameterization. We thus found important to let the community knows that such simulations already exist and can already be used for specific investigations. We added in the conclusions on lines 724-727 these considerations. Also we expanded the discussion about the TOA imbalance (see lines 450-461), now explaining our strategy to fix the energy imbalance (see reply to next comment).

b. Model drift: Fig 4 shows that TOA energy is adjusting rapidly over the course of this simulation. It would be nice to see a similar graphic for global-average surface temperature. Lack of discussion of what this drift means was conspicuously absent from the paper. Do you really think you could do a long simulation with this configuration? Do you have plans for fixing the drift?

As suggested by the reviewer, we added a panel of global-average surface temperature in Fig. 4. No we don't think that we can do a long simulation with this configuration (see our reply to previous comment), which we now acknowledge in the conclusions (lines 723-726). Yes we have a plan for fixing the drift. The TOA imbalance is mostly related to a preponderance of low clouds.  These arise when using the Smagorinsky scheme, which has a mixing cutoff. Our formulation sets the eddy diffusivities to zero if the Richardson number is greater than the eddy Prandtl number. This cutoff unrealistically inhibits mixing, both because of well known limitations of the Smagorinsky scheme in simulating the transition to turbulence (Porté-Agel et al., 2000), and because of a failure to incorporate the effect of moist processes.  As a result, over cold and moist surfaces, insufficient ventilation of the boundary layer occurs, causing moisture to build up and resulting in excessive low clouds. In ongoing experiments, we have explored adding a small amount of background mixing at interfaces between saturated and unsaturated layers where the equivalent potential temperature decreases upward, mimicking the effects of buoyancy reversal (Mellado, 2017). Low clouds respond sensitively to this background mixing, what provides a convenient control on their amount and on their influence on the top of the atmosphere energy budget. Ongoing work is

exploring theoretical justifications for the choice of the background mixing, but it may also be set empirically, as a way to provide a better representation of the statistics of low clouds. We added these considerations on lines 450-461.

3. I think a lot of the ocean analysis is naïve because it doesn't acknowledge that it takes the ocean a long time to drift away from its initial condition. Thus a lot of the analysis is probably more reflective of having an initial condition which looks like observations rather than that your ocean dynamics are working correctly. You can get a sense of initial condition versus equilibrated model bias by comparing the output from a coarse-resolution coupled run at initialization, after 1 year, and after 500 yrs. I bet most of your fields of interest look a lot more like the initial condition than the 500 yr value. I'm not sure this means you should throw out your ocean analysis, but I do think you need to clearly articulate the potential source of good skill.

We agree that we didn't acknowledge the slow dynamics of the ocean. We also agree that we cannot say at the outset if our ocean dynamics are working correctly given the length of the simulation. Having said this, first we think it is important to show that there is no obvious bias in the ocean state as, as already said, one could already investigates interesting scientific questions just based on one year of simulation. Despite the long experience of running ICON at low resolution in a coupled mode (Jungclaus et al., 2021), that the ocean works and couples correctly to the atmosphere is not given. In fact, several bugs were actually discovered during the development phase, bugs related to the momentum coupling between ocean and atmosphere. We thus added both the note of caution related to the slow ocean dynamics and the motivation for still validating the ocean at the beginning of section 4.1 on lines 423-431.  Second the ocean went through a nearly 85-year spin-up (albeit with 10-km grid spacing) and another 10 years of spin-up at 5 km. Going through this long spin-up also helped identifying issues in the ocean model. We added this on line 416. Third, the statistics considered for the validation of the ocean were salinity (Fig. 7), coupling (Fig. 8), barotropic streamfunction and water transport (Fig. 9), wind work (Fig. 10). Except for Fig. 9, all the other statistics are statistics that should respond fast when coupled to the atmosphere. Looking at Fig. 7 gives us some confidence that the biases that we see are not just a reflection of the initial state. As discussed previously in the paper, salinity biases arise at the mouth of big rivers, because we neglect river discharge, and, in the tropics, are consistent with precipitation biases. Hence, being related to the absence of river discharge and to the simulated precipitation pattern, the pattern of the salinity bias in the tropics is distinct from the one at the end of the spin-up period. This is confirmed by Fig. R1 below where we show the salinity bias in the last full month of the spin-up (December 2019) and the same month in G_AO_5km (December 2020). Except for the salinity bias in the Arctic, the pattern looks distinct. To clarify this point, we updated the discussion of Fig. 7 in the text, now mentioning which biases are inherited from the spin-up and which not, see lines 517 and 521-523. We did a similar analysis for the barotropic streamfunction and water transport (Fig. 9) by comparing values from the spin-up and from G_AO_5km. G_AO_5km

systematically shows weaker transport except in the Bering Strait (see updated Table in Fig. 9) and these smaller values are out of one standard deviation in all passages  but the Indonesian Throughflow and the Mozambique Channel. Except for the Florida Bahamas Strait, the weaker transport of G_AO_5km is in better agreement with observations. The weaker transport is consistent with weaker wind stress in G_AO_5km compared to the spin-up simulation, also expressed in a weaker barotropic streamfunction. These additional considerations have been added on lines 540-544 (together with the updated Fig. 9). Hence this supplementary analysis shows that the coupling leads to systematic differences to the uncoupled spin-up.

4. I'd like to hear more about conservation properties of the model. If I understand correctly, you've designed your schemes to have decent conservation properties and don't have a mass or energy fixer. I'd really like to see a plot of the unexplained global-average water and energy leak over time. This seems to me like it could be a huge problem for your multi-decadal simulation aspirations.

We don't have a water leak, water is conserved in the model, as mentioned on line 512. We have however several areas in the model, where energy is not conserved. The dynamical core unphysically extracts energy from the flow, at an amount of about 8 W m$^{-2}$, and precipitation is an unphysical source of energy. The former has been documented by Gassmann (2013). The latter arises because hydrometeors are assumed to have the temperature of the cell in which they are found. Because this assumption neglects the cooling of the air that accompanies the precipitation through a stratified atmosphere, it acts as an internal energy source, roughly (and coincidentally) of about equal magnitude to the dynamic sink. This explains why these energy leaks, which are also present in ICON-ESM, were not discovered previously. Moreover, minor energy leaks related to phase changes in the constant volume grid not conserving internal energy as well as to an inconsistent formulation of the turbulent fluxes have been discovered. Fixes for all of these problems have been identified and are being implemented. These considerations have been added on lines 462-470.

**Grammar, spelling, and details:**
1. ~L130: it seemed odd this paragraph doesn't include citations for readers to find out more about each of these component models, but then I realized that you go into a lot more detail about each component in following sections. If convenient, citations to the overview papers on each component model would be useful here. If not, please note that details about these models are given in section 2.

As the components don't have necessarily one key paper describing them, we prefer keeping the citations in the following sections, but we added at the beginning of this paragraph that details about the components are given in section 2 (see line 131).

2. L135: You say here that the atmosphere can only be run in uniform global or regional modes, but Fig 2 and elsewhere in the text talks about nesting, which seems like it uses several different resolutions in one run. The text also seems to imply that nesting is only available in regional simulations, which seems odd. Why can't you do nested regions inside a global run? Also, I think of telescoping as identical to nesting: you divide each tile of a coarse outer grid into finer but uniform grid cells and run a regional version of the model on this patch of fine-resolution cells. This fits with the idea of a telescope extending in a few discrete segments rather than continuously deforming to extend and retract. I think the way your ocean model works is that resolution is allowed to vary smoothly throughout the domain.

We see the nesting approach in the atmosphere and in the ocean as being different. As pointed out by the reviewer, in the ocean, the grid is allowed to be non-uniform and thus to vary smoothly throughout the domain in one simulation. This is not the case in the atmosphere where one simulation can only use a uniform grid and higher resolution is achieved by combining multiple simulations with distinct grid spacings, simulations which communicate through the boundaries of each respective domain. We clarified the text accordingly on lines 138-142.

3. Section 2.1: It would be good to mention in this section that aerosols are prescribed. You say this on line 303 in the I/O section, but readers will expect to hear about aerosol treatment in the atmos description.

We added this information on line 179.

4. Fig 3: Wow, this plot is cool – it has so much info. I'd prefer if dynamics and transport dots had their own line in the legend since closed and open dots obviously mean something uniquely different. Also, maybe add titles for the left and right columns of the legend since left is atm and right is ocean?

We split dynamics and transport on two lines in the legend (see new Fig. 3). We decided not to add titles for the left and right columns since in principle this information can be derived from the ICON-A, ICON-L, ICON-O labelling on the right of the plot, also given the fact that the figure is already quite busy with text.

5. Is land seen as just another atm process? It seemed odd that some components coupled via YAC but land doesn't. A sentence explaining why would be useful.

JSBACH is coupled implicitly to the atmosphere, and hence is tightly tied to it, something that doesn't work with YAC. The reason for using implicit coupling was that, at the time of development, many years ago, low-resolution simulations were in focus and there were stability concerns if using explicit coupling. For ICON-Sapphire, we are now working on rewriting the interface between the atmosphere and the land and are now using explicit coupling. In that case, one could actually use YAC, something that we might do in the future as this would allow JSBACH to be run on a different horizontal grid than the atmosphere. We don't want to go into all these details in the text, as those issues are not fully settled yet, but added that because of the implicit coupling, the land is not coupled via YAC (except for discharge), see line 135.

6. I thought Bjorn said at Pan-GASS that you have another turbulence option (Deardorff?) which was unintentionally acting as a shallow convection scheme. Is that worth mentioning here in the same spirit as you mention 2 moment microphysics and RTE-RRTMGP but don't use it?

We have indeed another turbulence option, which is the TTE scheme that we inherited from the ICON-ESM model. The TTE scheme was not active in the simulations presented in this paper. We are not mentioning the TTE scheme as we only would like to have one turbulence scheme in the future, and this will be a slightly modified version of the Smagorinsky scheme. In contrast, we will likely keep the two microphysics schemes (one moment and two moment). For the radiation, we are also only keeping one scheme, RTE-RRTMGP, but since we did the simulations with PSrad, we had to mention that scheme. We added a sentence in the text to make clear that PSrad will not be part of future releases of ICON-Sapphire, see lines 184-185.

7. Does ICON include horizontal turbulent mixing, or just vertical? It seems like horizontal mixing will be important at the hectometer scales you run at.

Yes, the reason for using Smagorinsky in place of the TTE scheme was that Smagorinsky also performs horizontal turbulent mixing. We added this information on lines 200-201.

8. L195: citation for Richtmyer and Morton numerical scheme?

The reference is Richtmyer and Morton, 1967: Difference methods for initial-value problems. We added the reference on line 212.

9. L246 – unclear what "latter" refers to.

The sentence was unclear. We rewrote it, see lines 253-254.

10. L278: What do "processes" refer to here? I think you mean the land model, atmosphere model, etc. I tend to think of these as "component models" with "processes" being particular physics schemes within a component... but that might be idiosyncratic of me. The concept of "neighboring processes" seems odd since processes have no spatial relationship to each other. I think you mean the process called before or after in sequential time splitting?

Here, we meant MPI processes and compute domains (local partitions of the horizontal grid) rather than physical processes. We rephrased this sentence to avoid misunderstandings, see lines 289-291.

11. L281: Doesn't the atmosphere just compute wind stress and provides that to whatever land model wants it? The way it's written, it sounds like the atmosphere. provides a different wind stress to sea ice versus ocean.

The atmosphere indeed provides a different wind stress over sea ice and over ocean. The atmosphere computes the wind stress for each of the surface tiles

separately. Although the velocity over ocean and sea ice is the same, the drag coefficient is not so that the wind stress is also different. We clarified the sentence, see lines 292-293.

12. L286: I think you should delete "and" between "wind" and "vectors"
We rephrased simply to "The interpolation of the wind is done", see line 297.

13. L298: It would be handy to point out that 30" is equal to ~900m at the equator.
Added (line 309).

14. L325: "single-precision 32-bit float arrays are now kept in memory". I think you mean that "output is now stored in single instead of double precision, reducing memory requirements by a factor of 2". "kept in memory" sounds like you mean the data is kept in cache instead of slower-access disk and "single-precision 3d bit" is redundant.
Yes and no. The issue is that we store ICON output as 32-bit array, whereas CDO employs double precision for calculation. So before, when using CDO, we were transforming 32-bit data to double precision, which can significantly slow down the calculations for memory-intensive operations. Now we don't do this transformation anymore. We updated the text to clarify, see lines 337-339.

15. L175 – I'm confused how you obtain good performance on GPUs if you use PSrad for all calculations in this paper and PSrad only runs on CPUs – I would have thought having some processes on CPU and others on GPU would result in excessive communication overhead and slow runs. Is radiation running in parallel with other atm processes? Or is it just that you call radiation so infrequently (every 15 min!) that it doesn't matter? I suspect you are forced to run radiation so infrequently precisely because it is on CPU.
We apologize, this was confusing indeed. All the simulations presented in Table 1 use PSrad but the experiments performed by Giorgetta et al. (2022), where the performance on GPU was assessed, employed RTE-RRTMGP. We clarified the text, see lines 371-374.

16. L339: I think 40 TB/month is for *5 km dx*. It would be useful to point this out and to also say how much storage space you're using for the 2.5 km grid (which I expect is 4x more = 120TB/month!).
Yes the 40 TB/month is for 5km dx and indeed, for the 2.5 km grid, we generated about 135 TB of output in a month. We added this on lines 349-351.

17. L369: "half productive" is bad grammar.
We rephrased to less productive, see line 379.

18. Adding another panel to Fig 4 showing the annual cycle of global-average surface T would be useful.
We agree and did so (see new Fig. 4).

19. L447: observations misspelled.
Corrected (line 494).

20. L452: I'm unclear how negative TOA radiative imbalance would lead to enhanced radiative cooling. First, what's your sign convention? Does negative radiative imbalance imply that the planet is losing heat? If so, I would think enhanced radiative cooling would cause the radiative imbalance. But TOA radiative imbalance could also be caused by an excessive planetary albedo.
We apologize, our formulation was confusing and incorrect. We wanted to say that the too large precipitation amounts are indicative of a too strong radiative cooling and a too strong radiative cooling would be consistent with a negative TOA radiative imbalance. We clarified the text, see lines 497-499.

21. Fig 11: caption skips panels e and g.
We reformulated the caption to correct this (see new caption Fig. 11).

22. Fig 12: panel a and e seem redundant.
We don't think so as through the shading of the rain in panel e, the small-scale structure of the salinity field is hard to recognize in panel e whereas it is nicely visible in panel a.

23. L554: "my" should be "may"
Corrected (line 606).

24. Fig 15: I thought sea breeze was a weak example of 2.5 km resolution since it is also captured pretty well at 25 km resolution. Also, this graphic would be a lot better using wind vectors rather than colors for just zonal wind.
We agree and since we already have many figures, we decided to remove this figure and just now shortly mention in the text that ICON-Sapphire can capture mesoscale circulations and their effects on convection (see lines 600-601).

25. L586: "latter" rather than "later"?
Corrected (line 639).

a) G_AO_5km

b) Spin-up

[Figure]

Fig. R1: Monthly mean (December) salinity bias (g kg$^{-1}$) in G_AO_5km and in the spin-up ocean simulation. Observations from PHC climatology.

---

## Referee Report (RR1)

2nd review of Hohenegger's ICON-Sapphire coupled overview paper

As I said in the previous review, I think this is a well written paper on an important topic. I'm very pleased by how the authors responded to the suggestions of both reviewers, which I think makes the paper even better. I'm particularly happy with the injection of comments on my big-picture questions (e.g. how does/doesn't km-scale resolution help, what can we learn from a 1 yr simulation, etc). I suggest the authors are given one more chance to make minor edits in response to the comments below, with the understanding that their next revision will be immediately accepted.

1.  Reviewer 1 L304 comment (L324 in new manuscript): Saying you didn't spin up the land because it was unclear how to do so felt a bit naïve to me, but maybe that just reflects different capabilities at different modeling centers. In E3SM we frequently spin up the land model either by running with the land model interacting with an atmosphere that's continually nudged to reanalysis for the period leading up to our target start date or by running the land model in standalone mode driven directly by atmospheric observations. These approaches are explained in the most recent CAPT overview paper: https://agupubs.onlinelibrary.wiley.com/doi/10.1002/2015MS000490 . For SCREAM, we do land-model standalone runs at the full resolution of the model because they're cheap. With nudging, we use full-resolution for the land model but coarser resolution for the atmosphere. Another approach would be just to interpolate land conditions from coarser resolution coupled simulations (though this can lead to issues with land/sea masks). I mention these approaches not necessarily for you to mention in the paper, but in case it helps you come up with a plan for future runs.

    More importantly, I didn't get the sense from Fig R1 that any soil layers in the tropics are equilibrated (except maybe the deepest layer, but that's the one we least expect to equilibrate fast!). Do you really think your "3 of 5 soil layers in the tropics are spun up" statement is accurate? To be clear, I don't think this detail affects the integrity of the paper in any way and I'm just pointing it out to make the paper as perfect as possible.

    In any case, I do think it would be worth articulating more clearly what "land not spun up" means in this case – it could mean that there's no moisture in the soil at all or that soil moisture is initialized to a single value everywhere, or that a spatially-varying climatology is used.

2.  Reviewer 1's comment on Fig 5 (Connection between TOA and large scale circulation): while the atmospheric general circulation responds rapidly to a given forcing change, SST changes slowly and the atmospheric general circulation will evolve in response to those slow changes. In that context, TOA energy imbalance will definitely cause sea level pressure and zonal winds to evolve if you ran your simulation longer. I don't think there's anything to change in your paper – adding the individual observed years in the graphic is nice and you've acknowledged now that the short length of your simulations

precludes definitive analysis of ocean-related variables. I just thought that the response to reviewer 1 wasn't correct.

3. L511 in new draft: it felt odd to me that you say there are 2 parallel bands of precipitation in the W Pacific instead of saying you seem to have a double ITCZ. It reminded me of describing a camel as a horse-like animal with 2 bumps on its back – an accurate description but more verbose than needed and harder for people to connect to. It's fine if you have a reason to avoid calling it a double ITCZ, but if not you might as well use standard terminology.

4. Reviewer 1 comment on L623: is the effect of mesoscale eddies on phytoplankton production in your high-res and coarser rest simulations in the same direction as found in Harrison et al 2018? If so, that's stronger evidence for your finding. If not, it suggests the difference may be due to chance.

5. Reviewer 2 comment 20: I agree that excessive precipitation implies too much radiative cooling *within the atmosphere*, but I disagree that excessive atmospheric radiative cooling implies a TOA radiative imbalance where more radiation is leaving the planet. My reasoning is that the heat capacity of the atmosphere is negligible compared to the surface, so TOA imbalances are more likely to reflect a surface that is too emissive or an atmosphere that reflects (rather than emits) too much energy.

---

## Author Response (AR2)

**Reply to reviewers' comments**

We thank both reviewers for their careful second review of our manuscript. Below are our responses to their technical edits/minor comments. The reviewers' comments are in blue, our responses in black.

**Reviewer 1**

The reviewer thanks the authors for their comprehensive addressing of the initial comments on the manuscript, I am happy that these were all addressed. I only have a few further detailed edits to suggest.

Thanks.

Detailed comments:

L105: are represented explicitly
Corrected (l.105)

L274: rehology --> rheology
Corrected (l. 271)

L376: I think the start of this sentence could be improved: "Such a throughput would be possible already now on the new petascale Europe's..", for example "Such a throughput would now be possible on petascale machines such as Europe's.."
The sentence was changed according to the reviewer's suggestion (l. 377).

Figure 4: The caption refers to a), b), but these labels are not on the figures. Also I think HadCRUT stands for "Hadley Centre/Climatic Research Unit Temperature".
Labels a) and b) were added to Figure 4 and HadCRUT was changed to Hadley Centre/Climatic Research Unit Temperature in the caption of Figure 4.

Figure 5: Similarly the caption refers to a), b) etc, but there are no labels.
There were actually a), b) etc labels on Figure 5, they are to be found on the lower right corner of each panel. We put the labels on the lower right corners as the upper left corners are already busy with the curves (especially on Fig. 5c).

L563: "potentially pinpointing to.." – would this be better as "potentially pointing to.."
Changed accordingly (l. 564)

L677: Altantic --> Atlantic
Corrected (l. 678)

**Reviewer 2**

2nd review of Hohenegger's ICON-Sapphire coupled overview paper

As I said in the previous review, I think this is a well written paper on an important topic. I'm very pleased by how the authors responded to the suggestions of both reviewers, which I think makes the paper even better. I'm particularly happy with the injection of comments on my bigpicture questions (e.g. how does/doesn't km-scale resolution help, what can we learn from a 1 yr simulation, etc). I suggest the authors are given one more chance to make minor edits in response to the comments below, with the understanding that their next revision will be immediately accepted.
Thanks.

1. Reviewer 1 L304 comment (L324 in new manuscript): Saying you didn't spin up the land because it was unclear how to do so felt a bit naïve to me, but maybe that just reflects different capabilities at different modeling centers. In E3SM we frequently spin up the land model either by running with the land model interacting with an atmosphere that's continually nudged to reanalysis for the period leading up to our target start date or by running the land model in standalone mode driven directly by atmospheric observations. These approaches are explained in the most recent CAPT overview paper: https://agupubs.onlinelibrary.wiley.com/doi/10.1002/2015MS000490 . For SCREAM, we do land-model standalone runs at the full resolution of the model because they're cheap. With nudging, we use full-resolution for the land model but coarser resolution for the atmosphere. Another approach would be just to interpolate land conditions from coarser resolution coupled simulations (though this can lead to issues with land/sea masks). I mention these approaches not necessarily for you to mention in the paper, but in case it helps you come up with a plan for future runs.

Maybe we should have added in our response that it is unclear how to spin up the land *in global climate storm-resolving simulations*. We added this nuance in the manuscript on lines 318-319. As far as we are aware, no study investigated the effect of soil moisture initialization in this type of simulations. We agree with the reviewer that starting from soil moisture conditions close to observations can improve the forecast skill for short-range simulations, as also known from storm-resolving weather forecasts, but it would be more important to know if the soil moisture initialization can affect the climate, either in terms of mean or pdf.

More importantly, I didn't get the sense from Fig R1 that any soil layers in the tropics are equilibrated (except maybe the deepest layer, but that's the one we least expect to equilibrate fast!). Do you really think your "3 of 5 soil layers in the tropics are spun up" statement is accurate? To be clear, I don't think this detail affects the integrity of the paper in any way and I'm just pointing it out to make the paper as perfect as possible. In any case, I do think it would be worth articulating more clearly what "land not spun up" means in this case – it could mean that there's no moisture in the soil at all or that soil moisture is initialized to a single value everywhere, or that a spatially-varying climatology is used.

Our assessment on whether a soil layer was spun up or not was based on an assessment of whether a persistent trend in soil moisture was visible, e.g. soil moisture decreasing throughout the simulation. In Fig. R1, it can for instance be seen that in the lowest two layers, the soil moisture keeps on increasing with time, even though not always at the same speed. Hence the soil is not spun up. In contrast, in the first layer, soil moisture decreases first, in January and February, before beginning oscillating. Hence, for that layer, we only interpret January and February as months where soil moisture is not spun up yet. And so on for the other layers. To clarify how we assessed the spin-up period, we added on line 321 "in the sense that the continuous drying of the soil since simulation start has stopped".

2. Reviewer 1's comment on Fig 5 (Connection between TOA and large scale circulation): while the atmospheric general circulation responds rapidly to a given forcing change, SST changes slowly and the atmospheric general circulation will evolve in response to those slow changes. In that context, TOA energy imbalance will definitely cause sea level pressure and zonal winds to evolve if you ran your simulation longer. I don't think there's anything to change in your paper – adding the individual observed years in the graphic is nice and you've acknowledged now that the short length of your simulations precludes definitive analysis of ocean-related variables. I just thought that the response to reviewer 1 wasn't correct.

We agree with the reviewer that we cannot rule out at the outset that there might not be a connection, but as also noted by the reviewer, we acknowledged this in our manuscript by adding the individual observed years in the graphic and by explicitly mentioning the limitation due to the length of our simulation.

3. L511 in new draft: it felt odd to me that you say there are 2 parallel bands of precipitation in the W Pacific instead of saying you seem to have a double ITCZ. It reminded me of describing a camel as a horse-like animal with 2 bumps on its back – an accurate description but more verbose than needed and harder for people to connect to. It's fine if you have a reason to avoid calling it a double ITCZ, but if not you might as well use standard terminology.

We rephrased to "This leads to the formation of a double ITCZ over the western Pacific with two parallel precipitation bands, where the southern band is too zonal, reaching too far east and remaining south of the equator even during boreal summer" (l. 491-492).

4. Reviewer 1 comment on L623: is the effect of mesoscale eddies on phytoplankton production in your high-res and coarser rest simulations in the same direction as found in Harrison et al 2018? If so, that's stronger evidence for your finding. If not, it suggests the difference may be due to chance.

We totally agree with the interpretation of the reviewer and actually the effect of mesoscale eddies on ocean biogeochemistry and especially carbon is the topic of a current PhD study, which was also one of the main reasons for conducting the ocean-only high-resolution simulation with biogeochemistry. Hence we prefer not touching upon this topic in this paper.

5. Reviewer 2 comment 20: I agree that excessive precipitation implies too much radiative cooling within the atmosphere, but I disagree that excessive atmospheric radiative cooling implies a TOA radiative imbalance where more radiation is leaving the planet. My reasoning is that the heat capacity of the atmosphere is negligible compared to the surface, so TOA imbalances are more likely to reflect a surface that is too emissive or an atmosphere that reflects (rather than emits) too much energy.

We decided to remove the corresponding sentence in the manuscript, which was "A too strong radiative cooling would be consistent with the fact that G_AO_5km is loosing too much heat compared to observations" as it was not precise enough and could lead to false conclusions. The negative net TOA imbalance in our simulation compared to observations is primarily due to too much shortwave reflection, as noted in the manuscript when discussing Fig. 1, but too much shortwave reflection would affect similarly the net TOA and surface radiation balance, meaning that it would not be visible in the atmospheric radiative cooling. Instead, an excessive atmospheric radiative cooling could be consistent with too little shortwave absorption in the atmosphere, a too cold Earth's surface or too much incoming longwave radiation at the surface.